# Mechanisms of cellular crosstalk in the gastric tumor microenvironment are mediated by YAP1 and STAT3

Pathum Thilakasiri[1,*], Ryan N O'Keefe[1,*], Sarah Q To[1], David Chisanga[1], Moritz F Eissmann[1], Annalisa LE Carli[1], Belinda Duscio[1], David Baloyan[1], Rhynelle S Dmello[1], David Williams[1,2], John Mariadason[1], Ashleigh R Poh[1], Bhupinder Pal[1], Benjamin T Kile[3], Joseph HA Vissers[4], Kieran F Harvey[4,5], Michael Buchert[1], Wei Shi[1], Matthias Ernst[1,†], Ashwini L Chand[1,†]

**Deregulation of the Hippo pathway is a driver for cancer progression and treatment resistance. In the context of gastric cancer, YAP1 is a biomarker for poor patient prognosis. Although genomic tumor profiling provides information of Hippo pathway activation, the present study demonstrates that inhibition of Yap1 activity has anti-tumor effects in gastric tumors driven by oncogenic mutations and inflammatory cytokines. We show that Yap1 is a key regulator of cell metabolism, proliferation, and immune responses in normal and neoplastic gastric epithelium. We propose that the Hippo pathway is targetable across gastric cancer subtypes and its therapeutic benefits are likely to be mediated by both cancer cell–intrinsic and –extrinsic mechanisms.**

## Introduction

The Hippo pathway is a highly conserved signalling pathway that regulates key cellular processes including organ size, cell fate, and tissue homeostasis. Furthermore, deregulation of the Hippo pathway is implicated in cancer development, treatment resistance, and metastasis (1, 2, 3). The main downstream effector protein of the Hippo pathway is Yes-associated protein 1 (YAP1) and its paralogue transcriptional coactivator with PDZ-binding motif (TAZ), which are coactivators of the TEA domain transcription factors (TEADs). The Hippo pathway controls YAP and TAZ activity by phosphorylating them and modulating the rate at which they shuttle between the cytoplasm and nucleus (4). YAP and TAZ hyperactivity have been shown to drive many hallmarks of cancer, such as cell proliferation, survival, and metastasis (1, 3).

Upstream proteins such as the tumor suppressor neurofibromin 2 (NF2 or Merlin) participates upstream of the Hippo pathway core kinase cassette (which includes the kinases LATS1/2 and MST1/2) by phosphorylating and inhibiting YAP1/TAZ activity. Genetic aberrations in different Hippo pathway proteins, and *YAP1* and *TAZ* gene amplifications have been identified in numerous cancer types including lung, gastric, and gynaecological cancers, melanoma, and head and neck squamous carcinomas (3, 5, 6, 7, 8, 9, 10, 11, 12). The loss of *NF2* leads to increased YAP1 nuclear localization and is strongly associated with poor prognosis in patients with meningioma, ependymoma, and mesothelioma (13). Although loss of function of *NF2* and/or gene amplifications in *YAP1/TAZ* have not been identified in gastric cancer, *YAP1* expression is progressively elevated with advancing cancer stage when compared with expression patterns in normal gastric tissue (3). *YAP1* mRNA expression is up-regulated in gastric intestinal-type adenocarcinoma and gastric mixed adenocarcinoma subtypes and correlates to worse overall patient survival (11, 14).

YAP1 is also activated in cancers via dysregulated signalling of Src family kinases, growth factors, and cytokines. Specifically, glycoprotein 130 (gp130 or CD130)–dependent cytokine signalling is linked to increased Yap1 expression and activity and associated with *adenomatous polyposis coli*–driven colon cancer progression (15). In preclinical models of *Kras*-driven pancreatic ductal adenocarcinoma, tumor progression has been functionally linked to the concomitant activity of LIF-1, YAP1, and TAZ (16). Although these studies established a strong relationship between Yap1 and gp130 cytokine production in the establishment of pancreatic and colorectal tumors, the cellular mechanisms underpinning Yap1-dependent inflammatory processes in cancer remain undefined.

The role of inflammatory pathways in the development of gastric cancers is well documented (7, 17, 18). Our previous studies have demonstrated that amplified gp130-dependent IL-6, and to a

[1]Olivia Newton-John Cancer Research Institute and School of Cancer Medicine, La Trobe University, Heidelberg, Australia   [2]Department of Pathology, Austin Health, Heidelberg, Australia   [3]Faculty of Health and Medical Sciences, University of Adelaide, Adelaide, Australia   [4]Peter MacCallum Cancer Centre, Melbourne, Australia   [5]Department of Anatomy and Developmental Biology, and Biomedicine Discovery Institute, Monash University, Clayton, Australia

Correspondence: ashwini.chand@onjcri.org.au; matthias.ernst@onjcri.org.au
Joseph HA Vissers's present address is Centre for Cancer Research and Department of Clinical Pathology, University of Melbourne, Melbourne, Australia
*Pathum Thilakasiri and Ryan N O'Keefe contributed equally to this work
†Matthias Ernst and Ashwini L Chand are Co-Senior authors

greater extent, IL-11 signalling causes spontaneous neoplastic transformation of gastric tissue in homozygous $Gp130^{Y757F}$ ($Gp130^{FF}$) mice, comprising a knock-in mutation resulting in excessive, ligand-dependent Jak/Stat3 signalling (19, 20, 21, 22). Indeed, we have previously shown that conditional genetic ablation (20, 23) or therapeutic inhibition of Stat3 in tumor-bearing $Gp130^{FF}$ mice reduces their tumor burden in the stomach. In the current study, we sought to establish whether Yap1 activity was intrinsically associated with that of gp130/Stat3 signalling-dependent carcinogenesis and to clarify the underlying mechanisms that control gastric cancer initiation and development.

Our data demonstrate the parallel deregulation of the Hippo and gp130/Stat3 pathways in the $Gp130^{FF}$-dependent model of gastric adenocarcinoma. Conditional, Tff1:CreERT transgene-mediated $Yap1$ gene knockout in the gastric epithelium of $Gp130^{FF}$ mice reduced gastric tumor incidence. In animals with well-established disease, ablation of Yap1 activity significantly reduced tumor size to inhibit cancer progression. Transcriptomics profiling of gastric epithelial cells identified novel Yap1-dependent pathways that contribute to the malignant transformation of the gastric epithelium. We show that epithelial Yap1 drives gastric tumorigenesis through an autocrine, Stat3-dependent mechanism, and by modulating innate and adaptive immune responses.

Our findings highlight the importance of the crosstalk between the gastric mucosa and immune responses that drive tumor development in a proinflammatory environment. Furthermore, our data provide proof-of-principle evidence that inhibition of Yap1 activity presents a targeted therapeutic opportunity to control gastric cancer.

# Results

## Increased Yap1 expression positively correlates with heightened gp130/Stat3 signalling in a preclinical model of gastric cancer

Gp130, the $\beta$ receptor subunit is essential for signalling activation of IL-6 family cytokines contributing to neoplastic growth in various cancers (19, 20, 24, 25). In the present study, we used a specific activation model of the gp130/Stat3 pathway, the homozygous $Gp130^{FF}$ mutant mouse of gastric cancer, that is, driven primarily by inflammation. In this model, gastric adenomas spontaneously develop (Fig 1A and B). This is due to the enhanced activity of gp130/Stat3 signalling, associated with increased IL-6 and IL-11 levels measured in tumors, and elevated phosphorylation of Stat3 (Fig 1C–E) (19, 20). Interestingly, increased Yap1 levels were observed in developing and neoplastic gastric tumors when compared with normal gastric tissue. Subcellular localisation of Yap1 was assessed by immunohistochemistry in tumor sections of $Gp130^{FF}$ mice, revealing increased nuclear Yap1 localization indicative of increased transcriptionally active Yap1 (Fig 1F).

To identify $Yap1$-expressing cells, we used single-cell RNA-sequencing (scRNA-seq) to profile cells of gastric tissue from WT control mice and gastric tumors derived from $Gp130^{FF}$ mice (Fig S1A). We found that $Yap1$ was predominantly expressed in normal gastric epithelial and tumor epithelial cells, with lower expression also

found in cancer-associated fibroblasts and endothelial cells (Fig S1B). To provide additional insights into Yap1 activity, we evaluated expression patterns on $Taz$ and $Tead1$, key interacting partners of the active transcriptional Yap1 complex. High levels of $Taz$ and $Tead1$ were also co-expressed in gastric epithelial and tumor cells and in cancer-associated fibroblasts (Fig S1B). To further characterize the cellular origin of abundant Yap1 protein expression in the $Gp130^{FF}$ gastric tumors, we used well curated gene signatures to further analyse $Yap1$ mRNA expression in specialised gastric cell populations including parietal cells, chief cells, foveolar cells, mucous neck cells, metaplastic neck cells, neuroendocrine cells, immune cells, and tuft cells using scRNA-Seq (Fig S1C). $Yap1$ transcripts were abundant in parietal cells, foveolar cells, and metaplastic neck cells.

## Activation patterns of YAP1/TEAD1 and gp130/STAT3 across human gastric cancer subtypes

To further investigate the combined contribution of YAP/TEAD and gp130/STAT3 pathway activation in a clinical context, we evaluated publicly available, The Cancer Genome Atlas stomach adenocarcinoma datasets (TCGA-STAD) to establish expression patterns. Up-regulation of $YAP1$, $IL11$, $IL6$, and $STAT3$ mRNA transcript expression was observed in human gastric tumor samples when compared with normal gastric tissue (Fig 2A). Furthermore, when assessing members of the HIPPO pathway, positive correlation of $YAP1$ expression was observed only with $TEAD1$ (Table S1). Interestingly, $YAP1$ expression positively correlated with $IL6ST$, $IL11RA$, and $STAT3$ expression in the TGCA-STAD patient dataset. We therefore further analyzed expression patterns in gastric cancers stratified by molecular subtypes, EBV, microsatellite instability, genomic stable (GS), and chromosomal instability. Surprisingly, expression of $YAP1$, $STAT3$, $IL11$, or $IL6$ mRNA transcripts remained comparable across the molecular subtypes of gastric cancer (Fig S2). However, $TEAD1$ expression was highest in GS tumors and significantly higher in chromosomal instability tumors when compared with EBV and microsatellite instability gastric cancer subtypes (Fig S2). Interestingly, expression of the gene encoding the IL-11 receptor ($IL11RA$) and gp130 ($IL6ST$) was highest in GS subtype.

The cells of the tumor microenvironment (TME) influence the growth and metastatic characteristics of cancer and can provide rationale for treatment selection with both targeted therapies and immune checkpoint inhibitors (26). The classification of TME by transcriptomic analysis of >10,000 cancer patients identified four distinct TME subtypes as conserved patterns across 20 different cancers: immune-enriched non-fibrotic (IE), where markers for anti-tumor immune responses are highly expressed; immune-enriched and fibrotic (IE/F); fibrotic (F), and immune desert (D) (26). Analysis of these TME subtype maps showed JAK/STAT activation signatures as highest in IE and IE/F subtypes (26). We used this dataset to evaluate the expression patterns of $YAP1$, $STAT3$, $IL11$, $IL6$, $TEAD1$, $IL6ST$, and $IL11RA$ in gastric cancer patients within the four TME subtypes (Fig S2). As with the gastric tumor subtypes, expression of $TEAD1$, but not that of $YAP1$ or $STAT3$, varied across the TME subtypes. We found that $TEAD1$, $IL6ST$, $IL11$, and $IL11RA$ expression levels were highest in the IE/F and F subtypes of TME, suggesting a positive correlation of the expression across the TME

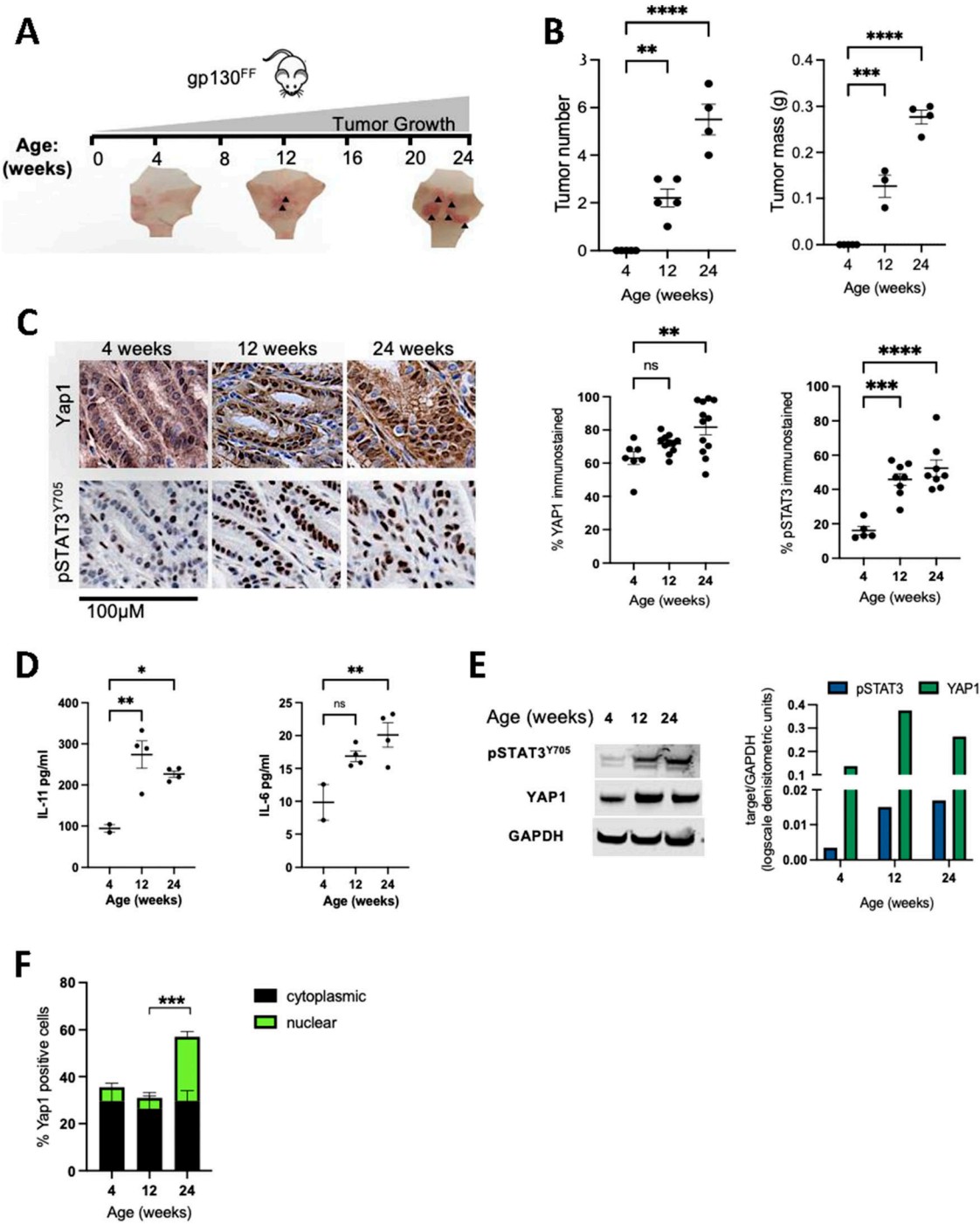

**Figure 1. Regulation of Yap1 expression in the *Gp130^FF* model of gastric cancer.**
**(A)** Tumor development timeline in the mutant *Gp130^Y757F* receptor model (*Gp130^FF*) for spontaneous gastric cancer growth. **(B)** Gastric tumor numbers and weights in homozygous *Gp130^FF* mice from 4 wk of age. Each data point depicts an individual animal, mean ± SEM from (n = 3–5 mice per cohort), **P < 0.001, ****P < 0.0001, one-way ANOVA with Turkey's multiple comparison test. **(C)** Representative Yap1 and phospho-Stat3 immunohistochemistry in gastric tumors from 4, 12, and 24 wk-old *Gp130^FF* mice and associated quantification. **(D)** Relative protein levels of the gp130 cytokines, IL-11 and IL-6 measured by ELISA. Each data point depicts a tumor derived from an individual animal. Data are mean ± SEM from (n = 2–4 mice per cohort), *P < 0.05, **P < 0.01, one-way ANOVA. **(E)** Western blot of phospho-Stat3 and Yap1 and corresponding densitometric quantification relative to GAPDH, in tumor lysates from *Gp130^FF* mice at different ages. **(F)** Quantitation of nuclear and cytoplasmic Yap1 immunostaining in tumors from *Gp130^FF* mice.

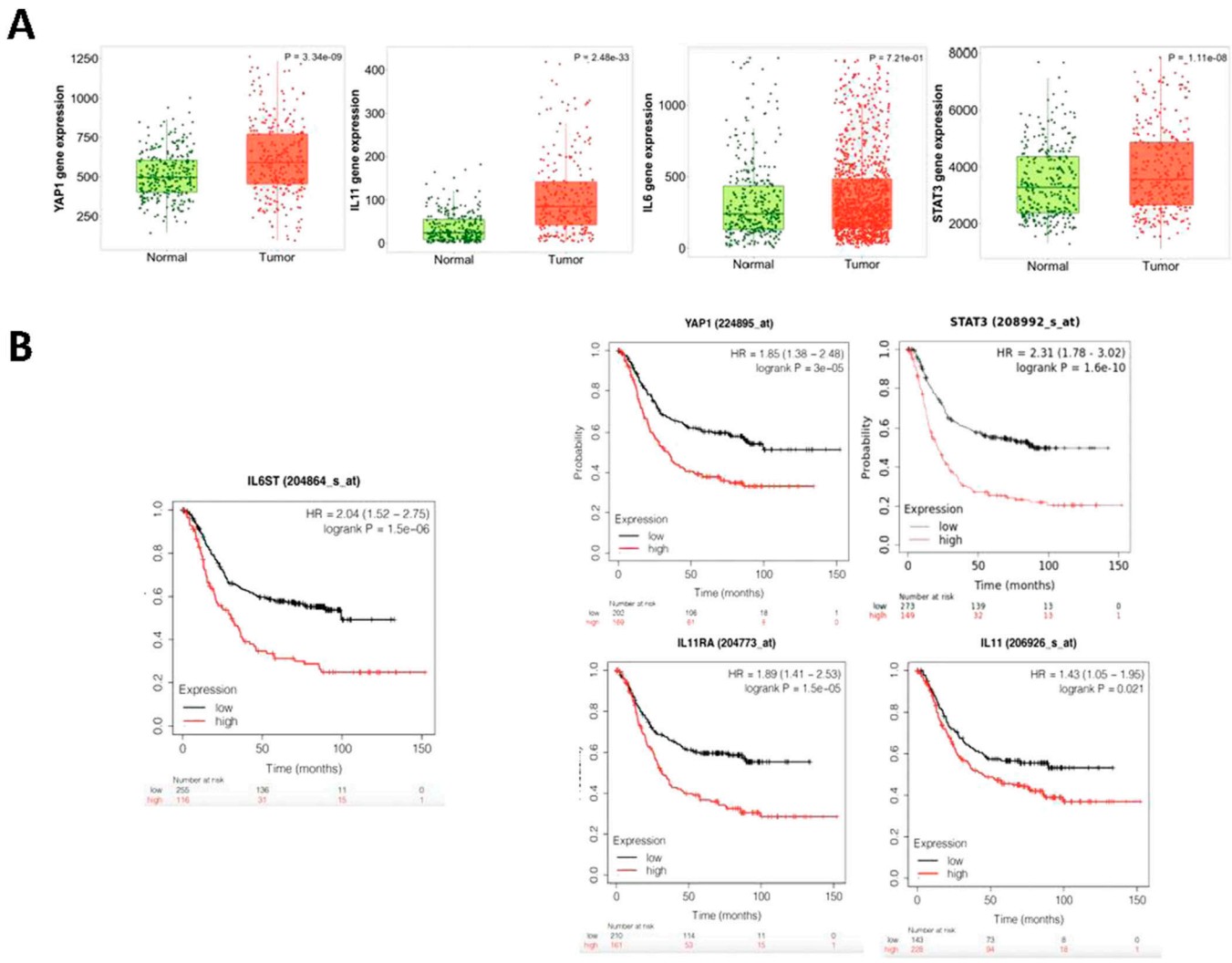

**Figure 2. Correlation between IL-11/STAT3 signalling components and YAP1 expression across human gastric cancer patients.**
**(A)** Box plots of gene expression of *YAP1* and *IL-11* in paired gastric tumor and adjacent normal human tissues in clinical data obtained from The Cancer Genome Atlas. *P*-values shown are derived from the Wilcoxon signed–ranked test. **(B)** Kaplan–Meier survival plots showing the association between gene expression levels of *IL6ST, YAP1, IL-11, IL11RA* with overall survival (OS) in all gastric cancer patients from all stages. Patient numbers, hazard ratio (HR), and the Mantel–Cox log-rank *P*-value are shown.

subtypes, and especially in tumors with high fibrotic stromal content and immune infiltration. Indeed, TME subtypes exhibiting fibrosis or a mesenchymal subtype despite high immune infiltrate are unresponsive to pembrolizumab (27), suggesting that therapeutic targeting the Hippo/Yap1 or STAT3 pathways could potentially enhance anti-tumor immune responses. To inform whether the above transcriptional observations impacted on overall outcome of gastric cancer patients, we assessed mRNA expression patterns of *YAP1* and components of IL-11/gp130/STAT3 signalling pathway against overall survival of patients within the TCGA-STAD dataset. We observed that high levels of *YAP1, STAT3, IL6ST, IL11RA,* and *IL11* correlated with significantly worse patient survival probability (Fig 2B). Collectively, our analysis of transcriptomics data from human gastric cancer patients and data from our in vivo gastric cancer model demonstrate the concomitant elevation of *YAP1* and receptor components of the gp130/STAT3 signalling pathway during gastric cancer development.

## Inhibition of Yap1 activity causes reduction in IL-11–dependent tumor growth in *Gp130^FF* mice

To assess the impact of impaired Yap1 expression in gastric tumors, we generated a *Gp130^FF*; *Tff1:CreERT2*; *Yap1^fl/fl* compound mutant mice and used tamoxifen-induced activation of Cre-recombinase to delete exon 3 of the *Yap1* gene, specifically in gastric epithelial cells in the mucosal layer of the corresponding *Gp130^FF*; *Yap1^KO* mice (Fig S3A and B) (25). The Trefoil factor 1 (*Tff1*) gene regulatory elements of the *Tff1:CreERT2* BAC-transgenic construct confers expression to foveolar epithelial surface cells of the gastric mucosa (28, 29) (Fig S1C) as the site of gastric cancer development. We first explored the function of Yap1 in well-established gastric tumors and administered tamoxifen to 16 wk-old when 100% of treatment naïve *Gp130^FF* mice exhibited large gastric lesions (Fig 3A). 4 wk following tamoxifen-induced activation of Cre, tumor burden was compared between the resulting *Gp130^FF*; *Yap1^KO* mice with Yap1-deficient

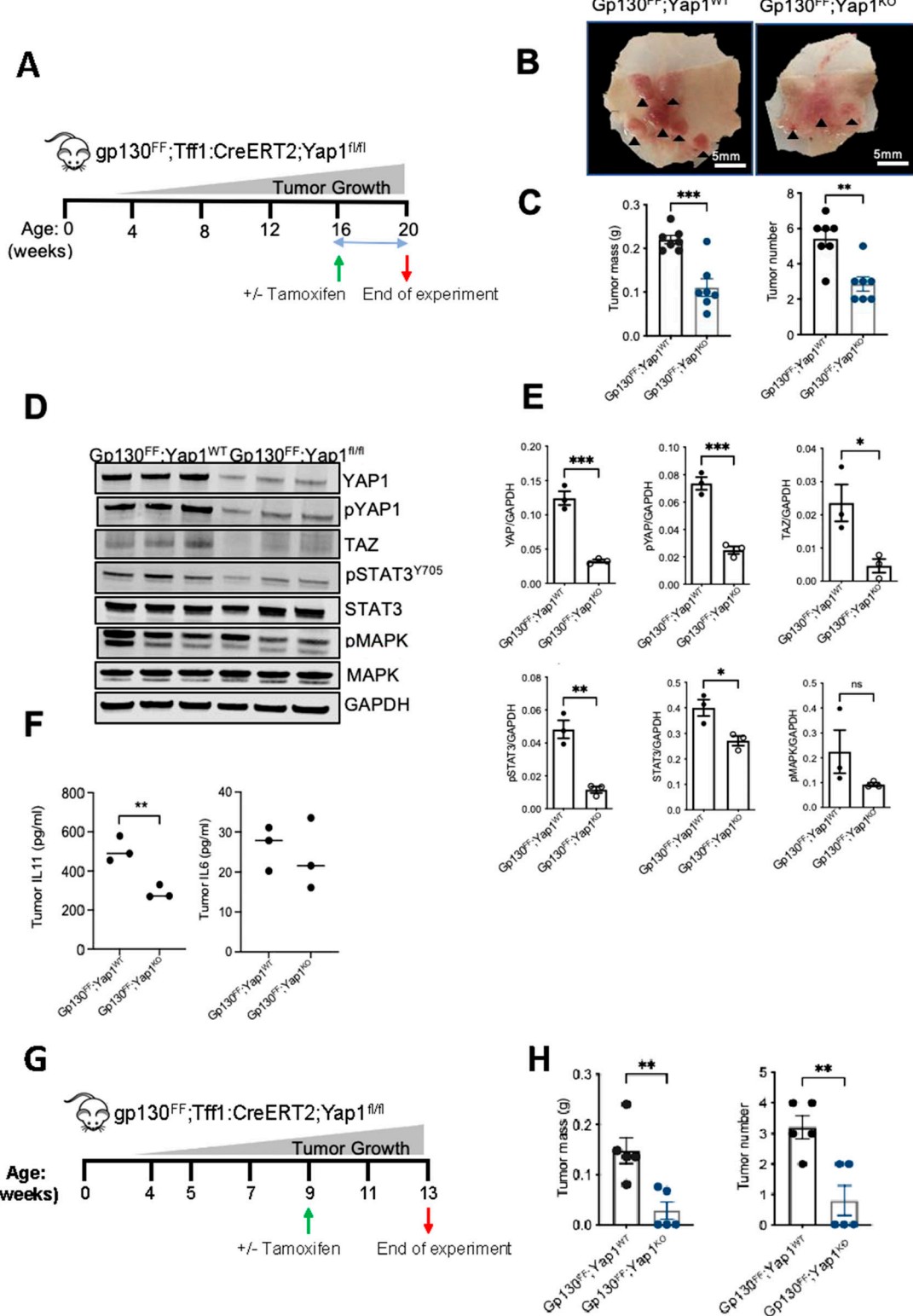

**Figure 3. Genetic ablation of Yap1 in gastric epithelial cells reduces tumor burden.**
**(A, B, C, D, E, F)** Schematic representation of experimental design to assess contribution of Yap1 to tumor progression assessed in (B, C, D, E, F). Yap1 is deleted specifically from the gastric epithelium because of tamoxifen activation of the CreERT2-recombinase at 16 wk of age when 100% of naïve *Gp130^FF* mice exhibit tumor burden of more than 0.15 *g* per stomach have established. **(B)** Representative whole mounts of stomach dissected along the outer curvature from *Gp130^FF*; *Yap1^WT* and *Gp130^FF*; *Yap1^KO* mice at 20 wk of age. Tumors are indicated by arrow heads, scale bar = 5 mm. **(B, C)** Total tumor mass and multiplicity determined in *Gp130^FF*; *Yap1^WT* and *Gp130^FF*; *Yap^fl/fl* mice as described in (B). Each data point represents an individual mouse, data are mean ± SEM, n = 7, **P < 0.001, ***P < 0.0001, unpaired *t* test.

gastric epithelium and their vehicle-treated Yap1-proficient $Gp130^{FF}$; $Tff1:CreERT2$; $Yap1^{fl/fl}$ littermates (referred to as $Gp130^{FF}$; $Yap1^{WT}$ mice Fig 3B). Gross morphological analysis of the dissected stomach revealed smaller tumors throughout the corpus and antrum of $Gp130^{FF}$; $Yap1^{KO}$ mice compared with $Gp130^{FF}$; $Yap1^{WT}$ littermates (Fig 3C). We confirmed that tamoxifen treatment did not affect tumor burden in $Tff1:CreERT2$ transgene-negative $Gp130^{FF}$ mice (Fig S4A and B).

Analysis of whole tumor protein lysates by Western blots demonstrated reduced levels of total and phosphorylated Yap1 and Stat3 in $Gp130^{FF}$; $Yap1^{KO}$ compared with $Gp130^{FF}$; $Yap1^{WT}$ mice (Fig 3D and E). Expression of Taz was significantly reduced as was activation of the MAPK pathways. In addition, we observed IL-11, but not IL-6 levels, were significantly reduced in $Gp130^{FF}$; $Yap1^{KO}$ derived tumor lysates (Fig 3F), consistent with our previous observation that IL-11 is a key player in gastric tumor development (21). From these observations, we infer that the reduction in phosphorylated Stat3 in tumors from $Gp130^{FF}$; $Yap1^{KO}$ mice is likely a result of lower IL-11 levels in Yap1-deficient tumors. These findings are supported by our observations in human gastric cancer samples, in particular of the GS subtype, of a positive correlation between components of the IL-11/gp130/STAT3 signalling pathway and YAP1/TEAD1 (Table S1 and Fig S2).

### Yap1 is important for inflammation-driven tumor initiation in the gastric epithelium of $Gp130^{FF}$ mice

To assess the effects of Yap1 on gastric tumor initiation, we also administered tamoxifen to $Gp130^{FF}$; $Tff1:CreERT2$; $Yap1^{fl/fl}$ mice at 9 wk of age (Fig 3G). 4 wk after Cre-recombinase induction, comparison of tumor mass and numbers showed significant reduction in tumor burden in the resulting $Gp130^{FF}$; $Yap1^{KO}$ mice compared with the vehicle-treated $Gp130^{FF}$; $Tff1:CreERT2$; $Yap1^{fl/fl}$ mice that had not received tamoxifen (referred to as to as $Gp130^{FF}$; $Yap1^{WT}$ Fig 3H). This suggests that Yap1 as a key driver of early events in inflammation-driven gastric tumor development in the $Gp130^{FF}$ mice.

### Yap1 is a key regulator of proliferative and metabolic pathways in gastric cancer

To identify transcriptional changes associated with $Yap1$ gene ablation, we isolated EpCAM$^+$ epithelial cells from gastric tumors obtained from 20 wk-old $Gp130^{FF}$; $Yap1^{WT}$ and $Gp130^{FF}$; $Yap1^{KO}$ mice for bulk RNASeq analysis (Fig 4A). We observed significant downregulation of genes associated with cell proliferation and metabolic pathways including glucose, lipid, cholesterol, and amino acid metabolism (Fig 4B; Table S2). Among the genes which were down-

regulated in $Gp130^{FF}$; $Yap1^{KO}$ mice, we not only detected $Yap1$ itself but also Sonic hedgehog ($Shh$) and ankyrin repeat domain 1 ($Ankrd1$) as prototypic target genes of the Hippo/Yap1 pathway (Fig 4C). Genes with reduced expression also related to mucosal function, included calcium-regulated non-lysosomal thiol-protease ($Capn8$), a marker for gastric surface mucus–producing cells, and Cilia- and flagella-associated protein 6 ($Cfap69$) as an abundantly expressed surface epithelial gene associated with sensory transduction. Reduced expression of genes regulating cell metabolism was also observed (Fig 4D–F). Kyoto Encyclopedia of Genes and Genomes (KEGG) pathway enrichment analysis of the downregulated genes showed significant changes in retinol, glutathione, steroid hormone, and amino acid metabolism (Fig 4D, Table S2).

We attributed the decreased tumor burden in $Gp130^{FF}$; $Yap1^{KO}$ mice to decreased cell proliferation as revealed by Ki67 immunohistochemistry, and we observed a concomitant increase in several apoptosis effectors including Bax, cell death activator ($Cidea$), pleckstrin homology–like domain family A member 3 ($Phlda3$), the E3 ubiquitin-protein ligase 186 ($Rnf186$), and tumor protein p53-inducible nuclear protein 1 ($Trp53inp1$) (Fig 4G and H). Furthermore, immunohistochemistry of tumor sections revealed significantly higher expression of cleaved caspase 3 in gastric tumors of $Gp130^{FF}$; $Yap1^{KO}$ mice when compared with $Gp130^{FF}$; $Yap1^{WT}$ mice (Fig 4I), confirming a clear up-regulation of multiple pathways that enhance apoptosis in response to $Yap1$ gene ablation.

### Loss of Yap1 induces an immune effector phenotype in gastric cancer

Interestingly, we also observed significant enrichment of genes associated with antigen processing and antigen presentation, phagosome function, cytokine signalling, NOD and RIG signalling, and cytosolic DNA-sensing pathways in tumors from $Gp130^{FF}$; $Yap1^{KO}$ compared with $Gp130^{FF}$; $Yap1^{WT}$ mice (Fig 5A and Table S2). The distinct up-regulation of genes associated with innate immune responses included interferon regulatory factors 7 and 9 ($Irf7$, $Irf9$) and MX dynamin-like GTPase 1 ($Mx1$) (Fig 5B and C). Expression of histocompatibility 2, Q region locus 6 ($H-2Q6$) and 7 ($H-2Q7$), and histocompatibility 2, T region locus 10 ($H-2T10$) expression were elevated. Although all three molecules are involved in antigen processing and presentation via MHC class I, H-2Q6 and H-2Q7 also enable T-cell receptor binding, suggesting altered instructions of the immune compartment by Yap1-deficient neoplastic gastric epithelium.

Given that our transcriptomics data demonstrated a strong up-regulation of immune-related genes, we used immunohistochemistry to quantify tumor-infiltrating immune cells. In tumors of $Gp130^{FF}$; $Yap1^{KO}$ we observed elevated numbers of CD3$^+$ T cells

**(B, D)** Western blot analysis of protein lysates from tumors collected in (B) for YAP1, phosphorylated YAP1 (pYAP1), TAZ, total and phosphorylated STAT3, and total MAPK and phosphorylated MAPK (pMAPK). Each lane corresponds to tumors pooled from an individual mouse, n = 3 mice per cohort shown. Blots were probed for GAPDH as a loading control. **(E)** Densitometric analysis of immunoblot bands for target antibodies. Each data point represents an individual mouse, data are mean ± SEM, n = 3, *$P < 0.05$, **$P < 0.001$, ***$P < 0.0001$, unpaired $t$ test. **(D, F)** Gp130 cytokines IL-11 and IL-6 levels in whole tumor lysates from (D) of $Gp130^{FF}$; $Yap1^{WT}$ and $Gp130^{FF}$; $Yap1^{KO}$ as measured by ELISA. Each data point represents an individual mouse, data are mean ± SEM, n = 3, **$P < 0.001$, unpaired $t$ test. **(G, H)** Schematic representation of experimental design to assess contribution of Yap1 to tumor initiation assessed in (H). **(H)** Tumor burden in 13-wk old mice following 4 wk after tamoxifen activation of the CreERT2-recombinase and yielding Yap1-deficient ($Gp130^{FF}$; $Yap1^{KO}$) or vehicle-treated cohorts yielding Yap1-proficient ($Gp130^{FF}$; $Yap1^{WT}$) mice. Data are mean ± SEM (n = 5 mice), **$P < 0.001$, $t$ test.

encompassing primarily CD4$^+$ helper T cells rather than CD8$^+$ cytotoxic T cells compared with $Gp130^{FF}$; $Yap1^{WT}$ mice (Fig 5D). Surprisingly, we also observed an increase in macrophages in Yap1-deficient gastric tumors.

To more accurately quantify the effects of Yap1 deletion in tumor cells on the tumor immune environment in vivo, we used flow cytometry to assess the abundance of various effector cells (Fig 6). When compared with $Gp130^{FF}$; $Yap1^{WT}$ tumors, those from $Gp130^{FF}$; $Yap1^{KO}$ mice showed increased proportion of infiltrating CD4$^+$ and NK cells, which were 20 times more abundant than CD8$^+$ T cells (Fig 6A, C, and E). We also found that CD4$^+$ and NK cells, rather than CD8$^+$ T cells, had increased expression of effector molecules granzyme B and perforin, associated with elevated expression of the exhaustion marker PD1 (Fig 6B, D, and F). Our FACS analysis also confirmed an increase in tumor-associated macrophages that we observed by immune histochemistry (Figs 5D and 6G). Collectively, our data suggest that altered Yap1 expression in tumor epithelium alters the composition and activity of the immune environment to possibly promote anti-tumor immune responses.

### Yap1 ablation suppresses proliferation of gastric tumor organoids

Although our analysis of $Gp130^{FF}$; $Yap1^{KO}$ mice clearly shows that tumor cell–intrinsic Yap1 expression impacts tumor burden, it remains unclear whether this is a direct effect reducing tumor growth or indirect effect arising from enhanced anti-tumor immune activity. We therefore derived $Gp130^{FF}$ tumor organoids and used CRISPR/Cas9-mediated gene editing to impair $Yap1$ expression by co-transfecting organoids with the tracrRNA containing an ATTO550 label and the guide crRNA that targets exon 1 of the mouse $Yap1$ gene (Fig S5A and B). Subsequent flow cytometric analysis for the ATO550-positive single cells suggested expression in a large majority of live cells, consistent with significant reduction of Yap1 at the mRNA transcript and protein level of organoids established from transfected cells (Fig 7A). The $Gp130^{FF}$; $Yap1^{KD}$ tumor organoids were smaller compared with $Gp130^{FF}$; $Yap1^{WT}$ organoids at the same time point in culture with reduced cell numbers (Fig 7B). Furthermore, the expression of the Hippo pathway target genes $Ankrd1$, $Cyr61$, $Amoltl2$, and $Taz$) was significantly reduced in $Gp130^{FF}$; $Yap1^{KD}$ tumor organoids (Fig 7C).

We then assessed expression of Yap1-dependent genes that we identified in the RNAseq analysis of tumors from $Gp130^{FF}$; $Yap1^{KO}$ mice, and which have prototypic functions in the associated KEGG cellular pathways (Fig S5C). In $Yap1^{KD}$; $Gp130^{FF}$ tumor organoids, we found reduced transcript levels for $Igsf9$ (marker for cell proliferation), $Cyp2c65$, (lipid and glucose metabolism), $Ptgr1$ and $Sult1c2$ (metabolite biosynthesis), and $Clca1$ (mucosal defence) when compared with $Yap1^{WT}$; $Gp130^{FF}$ tumor. By contrast, $Yap1^{KD}$; $Gp130^{FF}$ tumor organoids showed more prominent expression of $Chit1$ (marker for adaptive immune responses), $Cidea$ (apoptosis), and $Inf2$ (cell adhesion). Finally, we assessed whether Yap1-depletion in $Yap1^{KD}$; $Gp130^{FF}$ tumor organoids reduced expression of gp130 cytokines and found that indeed this effect was more pronounced on IL11 than IL6 (Fig 7D). Collectively, these observations confirm that the transcriptional changes observed in our RNAseq profiling of $Gp130^{FF}$; $Yap1^{KO}$ tumors arises as a direct consequence of reduced

epithelial Yap1 expression we observe in $Yap1^{KD}$; $Gp130^{FF}$ tumor organoids.

Finally, we confirmed tumor cell–intrinsic Yap1 expression as tumor-promoting mechanism outside a Stat3-driven model (i.e., $Gp130^{FF}$), also in a bona fide oncogene–driven gastric cancers model. Thus, we generated organoids derived gastric tumors arising in $Tff1:CreERT2$; $Kras^{G12D}$; $Pi3kCa^{H1027R}$; $Tp53^{R172H}$ (KPT) compound mutant mice in response to tamoxifen-dependent conversion of Kras and PI3K to their oncogenic isoform and simultaneous ablation of the p53 tumor suppressor gene (Eissmann, unpublished). Following CRISPR/Cas9-mediated impairment of $Yap1$ expression in KPT organoids, which we confirmed by reduced Yap1 mRNA and protein levels (Fig 7E). Importantly, we observed that Yap1-deficient KPT organoids were smaller than their Yap1-proficient KPT counterparts and this corresponded with reduced cell proliferation of Yap1-deficient KPT organoids (Fig 7F).

Collectively, our results obtained across Yap1 mutant $Gp130^{FF}$ and KPT tumor organoids suggests that inhibition of Yap1 activity confers suppression of tumors primarily via an IL11-mediated, tumor cell–intrinsic mechanism that possibly is further augmented by an enhanced anti-tumor immune response suggested from observations in $Gp130^{FF}$; $Yap1^{KO}$ mice. Thus, YAP1 provides a novel therapeutic target across gastric cancer subtypes that arise from various oncogenic triggers in humans.

## Discussion

Gastric cancer remains the second leading cause of cancer death worldwide, accounting for ~780,000 deaths per year (30). Increased gastric cancer incidence is observed in individuals aged younger than 50 yr (30, 31, 32). The heterogeneity of gastric cancer is characterised with molecular sub-classifications (7, 26) and has increasingly been correlated with patient outcomes (27, 33). Molecular classification, especially that of gastric TME subtypes has utility in the prediction of response to checkpoint inhibitors in gastric cancer patients (27). Although the analysis of patient tumor heterogeneity is extremely informative for personalization of treatments, these also highlight the need for additional targeted therapies to suppress the multitude of cancer pathways at play. This also highlights the need for functional studies that delve into cellular pathways important to gastric cancer development, in immune-competent, and relevant oncogenic preclinical model systems. In this vein, our study investigated the importance of the Hippo signalling pathway and its crosstalk with gp130/STAT3-mediated inflammatory pathways in gastric cancer development.

Our work previously demonstrated a strong causal link of gp130 proinflammatory cytokines IL-6, and more so, IL-11 in the development of gastric cancer (21, 22). There is emerging link between proinflammatory cytokine signalling and YAP1 function in the context of inflammatory diseases and cancer. In models of inflammatory bowel disease, the up-regulation of gp130 signalling mediated by elevated levels of tumor IL-6 and IL-11 leads to Yap1 activation as a mechanism of wound healing (34). This was mediated, in part, by the direct interaction of gp130 with tyrosine kinases SRC and YES, to facilitate YAP1-dependent actions in the

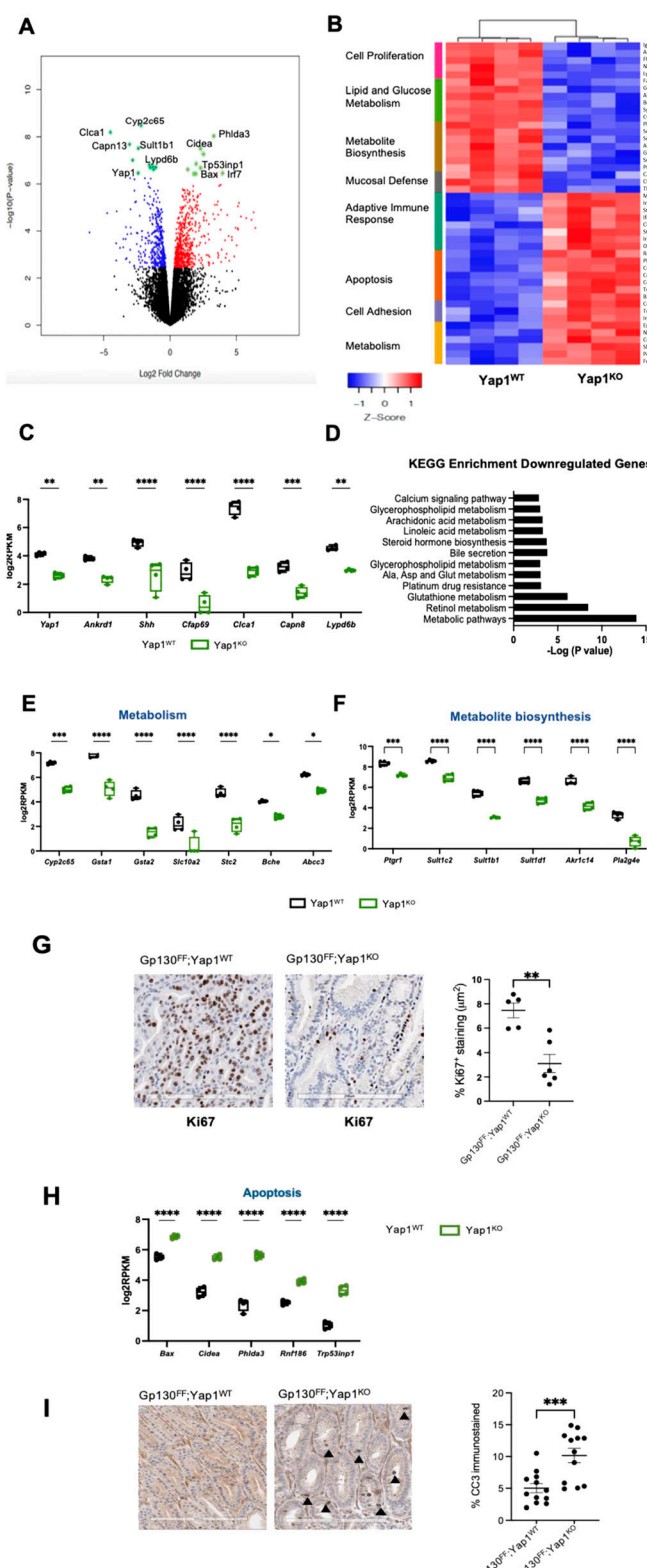

colonic epithelium (34). In colorectal cancer models driven by the loss of *adenomatous polyposis coli* gene, the up-regulation of gp130 expression leading to a heightened responsiveness to IL-6 and IL-11 also induces Yap1-dependent transcription (15). Although these studies have been correlative, the present study, in which we use a hyperactive gp130 mutant (*Gp130$^{FF}$*) gastric cancer mouse model, we demonstrate a causal relationship between gp130/IL-11 signalling and Yap1 function. Single-cell RNA-Seq analysis of murine gastric tumors demonstrated varied expression patterns of the Hippo pathway transcriptional complex members, namely, *Yap1*, *Taz*, and *Tead1*, in gastric tumor cells. The abundance of *Yap1* expression in normal epithelial and stem cell populations in WT gastric tissue, is indicative of its normal roles in gastric tissue homeostasis and tissue regeneration (35), as part of gut barrier function.

When we used scRNA-seq to profile the *Gp130$^{FF}$* gastric tumor lesions, *Yap1* expression was localised specialised, gastric parietal cells, foveolar cells, and in the tumor epithelial cells. There are several specialized epithelial cells that aid barrier function in the gastric mucosa. These specialised cells include chief cells, enteroendocrine cells, stem cells, and foveolar cells, all of which are also found in gastric tumors (36). Lower, but detectable expression was observed in chief cells, neuroendocrine cells, tuft cells, and immune cells, highlighting the need to assess the functional effects of Yap1 in these cells in more depth in future studies.

From the expression patterns of *Tff1* in the *Gp130$^{FF}$* gastric tumors, we can infer that the conditional deletion of Yap1 with the Tff1-driven Cre-recombinase targeted the parietal cells, chief cells,

**Figure 4. Transcriptomics analysis of Yap1-dependent pathways in gastric cancer development.**

**(A)** Volcano plot illustrating fold change of differentially regulated genes (x axis) versus statistical significance (y axis) in tumor epithelial cells isolated from 20-wk-old *Gp130$^{FF}$*; *Yap1$^{WT}$* and *Gp130$^{FF}$*; *Yap1$^{KO}$* mice. Tamoxifen was administered for

3 d in mice and tumor samples collected at the end of treatment for transcriptomics profiling. Top up-regulated and down-regulated genes are highlighted in green. **(B)** Heatmap of normalised log-2 rpkm RNA-seq data from 20-wk-old *Gp130$^{FF}$*; *Yap1$^{WT}$* and *Gp130$^{FF}$*; *Yap1$^{KO}$* mice. Unsupervised clustering of rows (genes) and columns (individual mice) for *Gp130$^{FF}$*; *Yap$^{WT}$* or *Gp130$^{FF}$*; *Yap$^{fl/fl}$* animal cohorts. Groups of genes related in biological function are indicated. **(C)** Box and whisker plots of individual mRNA transcript expression data for selected down-regulated genes, showing log$_2$RPKM values of tumor samples from *Gp130$^{FF}$*; *Yap1$^{WT}$* (black) and *Gp130$^{FF}$*; *Yap1$^{KO}$* (green) mice. Data points are values from individual mice (n = 4), ***P < 0.001, ****P < 0.0001 Sidak's multiple comparisons test. **(D)** Top most significantly down-regulated cellular pathways identified by Kyoto Encyclopedia of Genes and Genomes enrichment pathway analysis (y-axis) plotted against −log$_{10}$ P-value. **(E)** Box and whisker plots of individual mRNA expression data grouped by molecular markers for cell metabolic pathways, showing log$_2$RPKM values of tumor samples from *Gp130$^{FF}$*; *Yap1$^{WT}$* (black) and *Gp130$^{FF}$*; *Yap1$^{KO}$* (green) mice. Data points are values from individual mice (n = 4), ***P < 0.001, ****P < 0.0001 Sidak's multiple comparisons test. **(F)** Box and whisker plots of individual gene expression data grouped by regulation of metabolite synthesis, showing log$_2$RPKM values of tumor samples from *Gp130$^{FF}$*; *Yap1$^{WT}$* (black) and *Gp130$^{FF}$*; *Yap1$^{KO}$* (green) mice. Data points are values from individual mice (n = 4), ***P < 0.001, ****P < 0.0001 Sidak's multiple comparisons test. **(G)** Representative images of Ki67 immunohistochemistry and the quantification of Ki67 immunostaining of tumor sections from *Gp130$^{FF}$*; *Yap1$^{WT}$* and *Gp130$^{FF}$*; *Yap1$^{KO}$* mice. Data points are individual tumors from n = 5–6 mice, mean ± SEM, ***P < 0.001, unpaired *t* test. Scale bar = 200 *μ*m. **(H)** Box and whisker plots of individual gene expression data grouped by regulation of apoptosis, showing log$_2$RPKM values of tumor samples from *Gp130$^{FF}$*; *Yap1$^{WT}$* (black) and *Gp130$^{FF}$*; *Yap1$^{KO}$* (green) mice. Data points are values from individual mice (n = 4), ***P < 0.001, ****P < 0.0001 Sidak's multiple comparisons test. **(I)** Representative images of cleaved caspase 3 immunohistochemistry with positive stain highlighted with black arrowheads. Quantification of apoptosis by cleaved caspase 3 immunohistochemistry of tumor sections from *Gp130$^{FF}$*; *Yap1$^{WT}$* and *Gp130$^{FF}$*; *Yap1$^{KO}$* mice. Scale bar = 200 *μ*m. Data points are individual tumors from n = 11–12 mice, mean ± SEM, ***P < 0.001, unpaired *t* test.

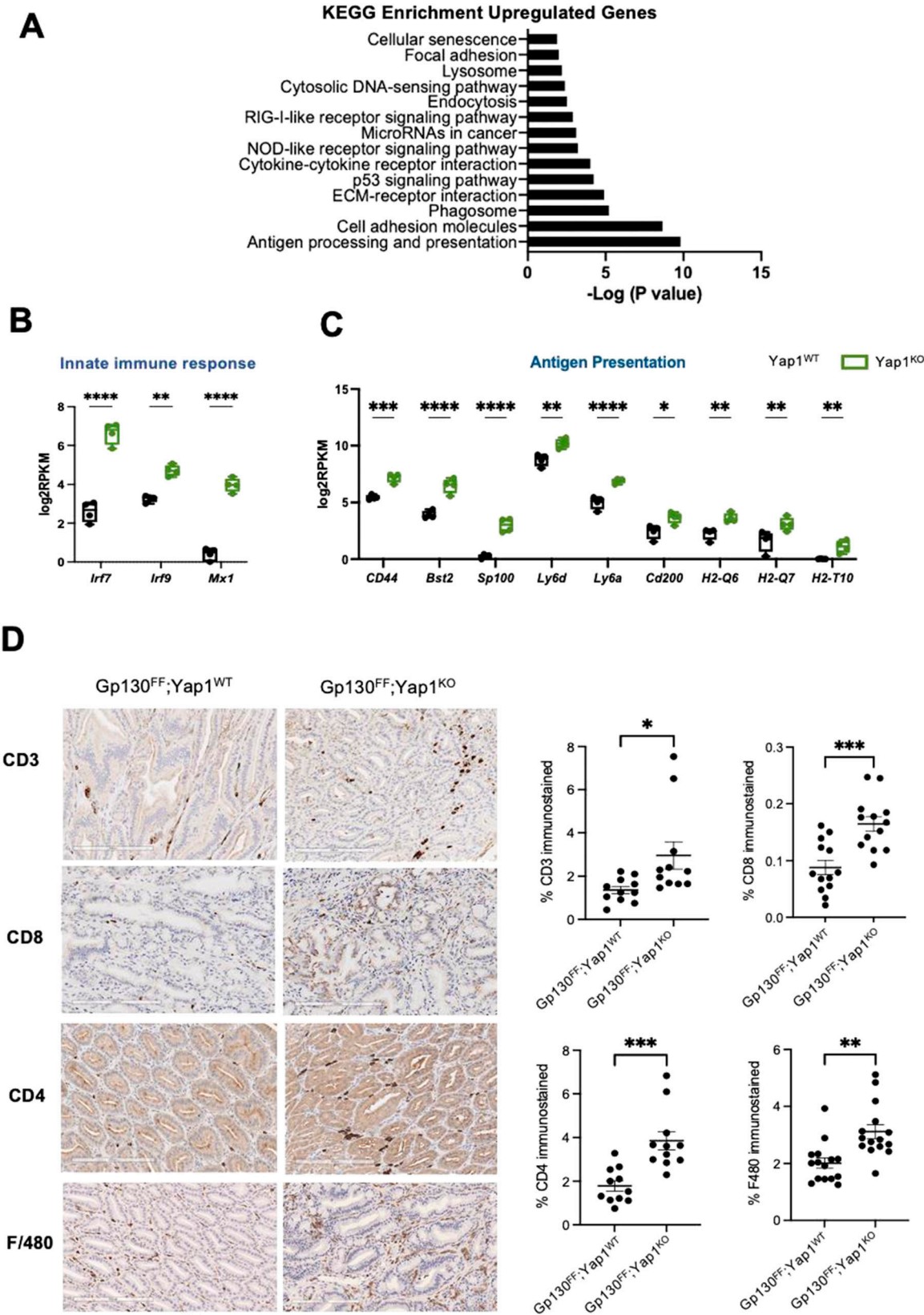

foveolar cells, tumor epithelial cells, mucous neck cells, tuft cells, and neuroendocrine cells, thereby inducing a potent anti-tumor effect. Pathway analysis of Yap1-dependent transcriptional gene signatures demonstrated down-regulation of proliferation and cell metabolic pathway genes and an up-regulation of genes that regulate apoptosis, antigen presentation, and innate immune responses. Gastric mucosal epithelial cells act as the first line of defence in orchestrating innate immune responses by providing vital crosstalk to modulate tissue resident macrophages and dendritic cell responses ([37], [38], [39]). In line with a proposed function of gastric mucosal cell in regulating T cell–mediated immune responses, we observed up-regulated expression of cell surface molecules including *Cd44*, *Lys6a* (*Sca1*), *Lys6d*, *Bst2*, histocompatibility 2, Q region locus 6 (*H2-Q6*, the mouse orthologue of MHC-1E), and 7 (*H2-Q7*) and histocompatibility 2, T region locus 10 (*H2-T10*) in Yap1-depleted gastric tumor cells.

When assessing the net effect of these transcriptomic changes on the immune landscape of tumors, we found that epithelial Yap1-deficiency increased infiltration and activation of CD4 and NK cells more than of CD8 cell. Indeed, flow cytometric characterisation of infiltrating CD4 and NK effector cells showed a selective increase in expression of granzyme B, perforin, and PD1 in these cells in tumors of *Gp130^FF*; *Yap1^KO* mice. The recruitment of NK cells by CD4⁺ T cells to mediate anti-tumor response is MHCI independent ([40]), suggesting multiple co-ordinated anti-tumor immune responses activated as a result of Yap1 depletion in the tumor epithelium. Although we have not functionally evaluated the contribution of specific immune effector cell populations, we note that ablation of IL-11 responsiveness of the tumor stroma resulted in increased activity of CD4⁺ cells, thereby increasing cytotoxic responses against tumor cells in addition to the proposed CD4⁺ T cell–mediated aid to cytotoxic CD8⁺ cells responses ([41], [42], [43]).

Gastric tumors in the *Gp130^FF* model are driven by IL-11 signalling, which we have documented previously through genetic ablation of the ligand-binding IL-11RA receptor subunit, or following therapeutic administration of the IL-11 antagonist IL-11-mutein ([21]). We have also demonstrated that pharmacological inhibition of IL-11 signalling with a small molecule gp130 inhibitor has potent anti-tumor effects in *Gp130^FF* mice and other models of gastrointestinal cancers ([22]). Here, we show that reduced expression of Yap1 in *Yap1^KD*; *Gp130^FF* tumor organoids reduces their capacity to produce IL-11, whereas Yap1 deletion reduces proliferation and survival of neoplastic cells in tumors of *Gp130^FF*; *Yap1^KO* mice. Thus, we surmise that the effect of Yap1 inhibition may be mediated through a cancer cell–intrinsic, auto-/paracrine IL-11–dependent mechanism, which may potentially be augmented by a concomitant anti-tumor effect emanating from reduced IL-11 signalling in CD4 effector cells ([42]).

The present study provides insights into novel link by which Yap1 signalling in tumor cells aids tumor progression through a direct, tumor cell–intrinsic, and a likely concomitant tumor cell–extrinsic mechanism involving the host's anti-tumor immune response. Our results suggest that these mechanisms may be controlled in parts by the capacity of Yap to promote in tumor cells expression of IL-11, possibly through a previously identified NF-κB–dependent mechanism ([10]). In turn, IL-11–dependent Stat3 signalling may be further augmented by Yap1's proposed capacity to heterodimerize with nuclear Stat3 to augment transcriptional responses ([44]). The resulting Stat3-dependent feed-forward signalling loop in tumor cells not only provides a therapeutic vulnerability for gastric tumors ([20], [23]) but also appears to help shaping aspects of the immune environment. This notion is consistent with the emerging role for IL-11 as a regulator for tissue repair and fibrosis where Yap1 also plays a prominent role ([45]). Our studies provide a strong rationale for the use of Hippo pathway inhibitors in the treatment of gastric cancers, and as a target that may enhance tumor immunogenicity to provide durable treatment responses.

## Materials and Methods

### Mouse models and experimental detail

All animal experiments were conducted in accordance to ethics protocols approved by Austin Health and La Trobe University Animal Ethics Committees. WT (C57BL/6 Ly5.2), *Gp130^Y757F* (*Gp130^FF*) mutant mice ([19]), *Yap1*^tm1a(KOMP)Mbp^ (MGI:4455424), the Tg (Tff1–CreERT2) mice ([25]), and the *Gp130^FF*; *Tff1:CreERT2*; *Yap1^fl/fl* mice were bred and maintained under pathogen-free conditions at Latrobe University animal facility and at the Austin BioResources Facility. To induce Cre-mediated exon deletion, a single daily interperitoneal (i.p.) injection of tamoxifen (50 mg/kg bodyweight) was administered over three consecutive days to *Gp130^FF*; *Tff1: CreERT2*; *Yap1^fl/fl* mice to yield Yap1-deficient cohorts (referred to as *Gp130^FF*; *Yap1^KO*), whereas vehicle-treatment of *Gp130^FF*; *Tff1: CreERT2*; *Yap1^fl/fl* mice yielded the Yap-proficient cohorts (referred to as *Gp130^FF*; *Yap1^WT*). All experiments were performed in age-matched and in male and female animals. Mice were euthanized by CO₂, and tissues were collected for further analysis including immunohistochemistry, Western blot, and qRT-PCR. Assessment of tumor burden and sample analysis was performed blinded by investigators. Gastric tumors from *Gp130^FF* mice and from Tff1-driven *Kras*^G12D^, *Pi3kCa*^H1027R^, and *Tp53*^R172H^ (KPT) transgenic mice were used to establish tumor-epithelial organoids (Eissmann et al, unpublished). Organoids were established and maintained in IntestiCult Organoid Growth Medium (StemCell Technologies) supplemented with IntestiCult OGM Mouse Supplement

---

**Figure 5. Inhibition of Yap1 function enhances anti-tumor immune responses.**
**(A)** Top most significantly up-regulated genes and canonical pathways as defined by Kyoto Encyclopedia of Genes and Genomes enrichment pathway analysis (y-axis) plotted against –log₁₀ *P*-value. **(B)** Box and whisker plots of individual gene expression data grouped by functional pathway, showing log₂RPKM values of tumors from *Gp130^FF*;*Yap1^WT* (black) and *Gp130^FF*;*Yap1^KO* (green) mice. Data points are values from individual mice (n = 4), *P < 0.05, **P < 0.01, ***P < 0.001, ****P < 0.0001 Sidak's multiple comparisons test. **(C)** Representative micrographs of immune cell infiltrates in tumors from *Gp130^FF*;*Yap1^WT* and *Gp130^FF*;*Yap1^KO* mice. **(C, D)** Quantification of immune cell infiltrates in tumors from *Gp130^FF*;*Yap1^WT* and *Gp130^FF*;*Yap1^KO* mice from (C). Scale bar = 200 μm. Data points are individual tumors from n = 11–15 animals, mean ± SEM, *P < 0.05, **P < 0.01, ***P < 0.001, ****P < 0.0001, unpaired *t* test.

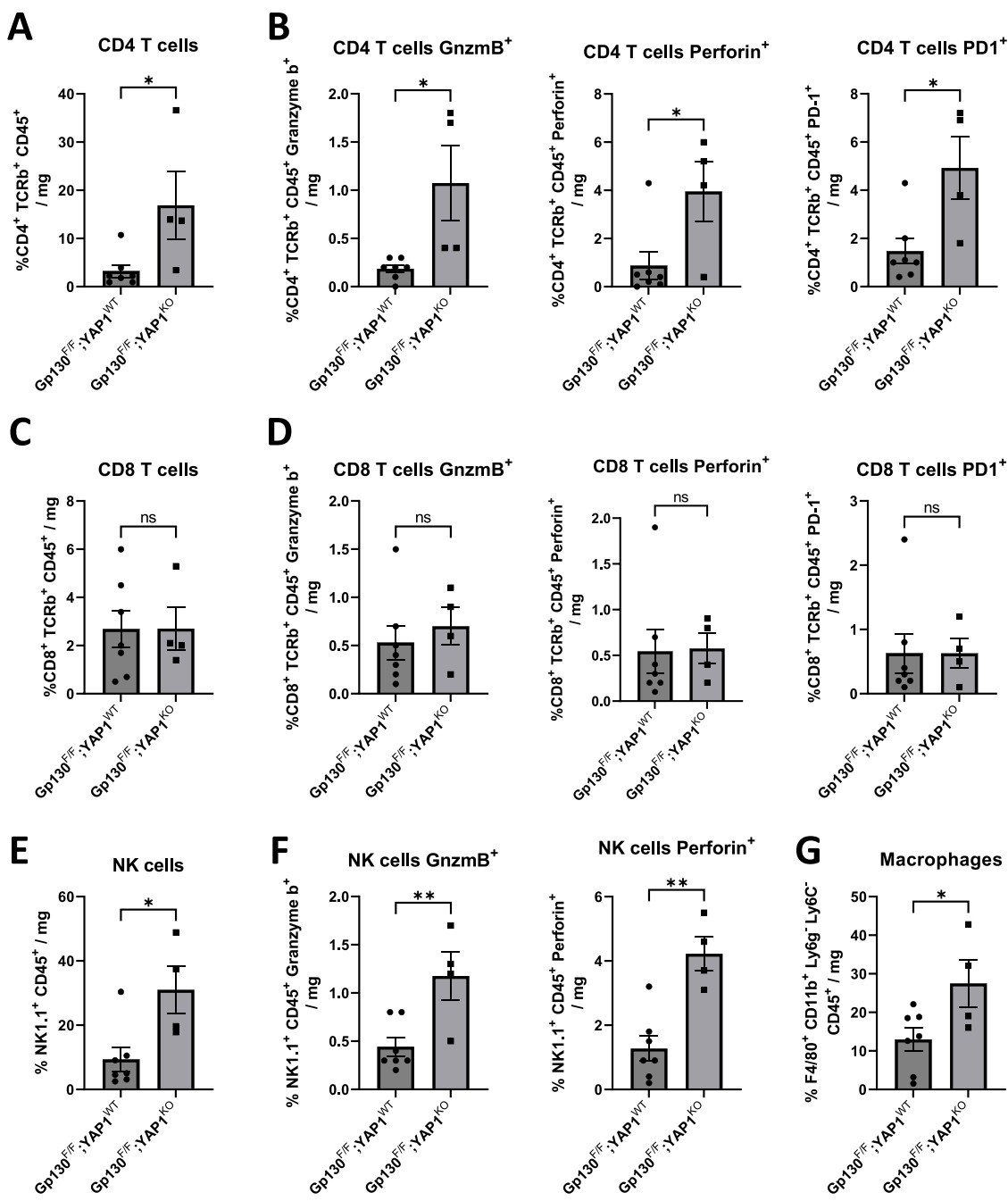

**Figure 6. Inhibition of Yap1 function enhances CD4 T cells and NK cell cytotoxic activity.**
Single-cell suspensions of gastric tumors were analyzed for tumor-infiltrating immune cells and corresponding activation markers using flow cytometry. **(A, C, E, G)** Contribution of total number of CD4[+] T cells (identified as CD45[+] TCRβ+ CD4[+] cells), CD8[+] T cells (identified as CD45[+] TCRβ+ CD8[+] cells), NK cells (identified as CD45[+] NK1.1), or macrophages (F4/80+ CD11b+ Ly6G− Ly6C−) expressed as number of cells per mg of gastric tumor. **(B, D, F)** Contribution of granzyme (Gnzmb+), Perforin+ or PD1+ expressing CD4[+] T cells, NK cells, or CD8[+] T cells (F) expressed as number of cells per mg of gastric tumor. T test (*$P < 0.05$). Each symbol represents a tumor derived from a separate mouse. For all panels, data are expressed as mean ± SEM.

1 and 2 (StemCell Technologies) as previously described (46). Organoids were passaged using Gentle Cell Dissociation Reagent (StemCell Technologies) to dissolve Matrigel and separate organoids into single-cell suspension. Organoids were cultured in 3D domes using Cultrex RGF BME, type 2 Matrigel (R&D Systems).

## Patient transcriptomics analysis

Patient transcriptomic data were obtained from The Cancer Genome Atlas Program data repository (https://www.cancer.gov/tcga). The HiSeq gene expression values (Log$_2$[RPKM + 1] of the STAD (n = TCGA patient cohort) was obtained from the Xena platform

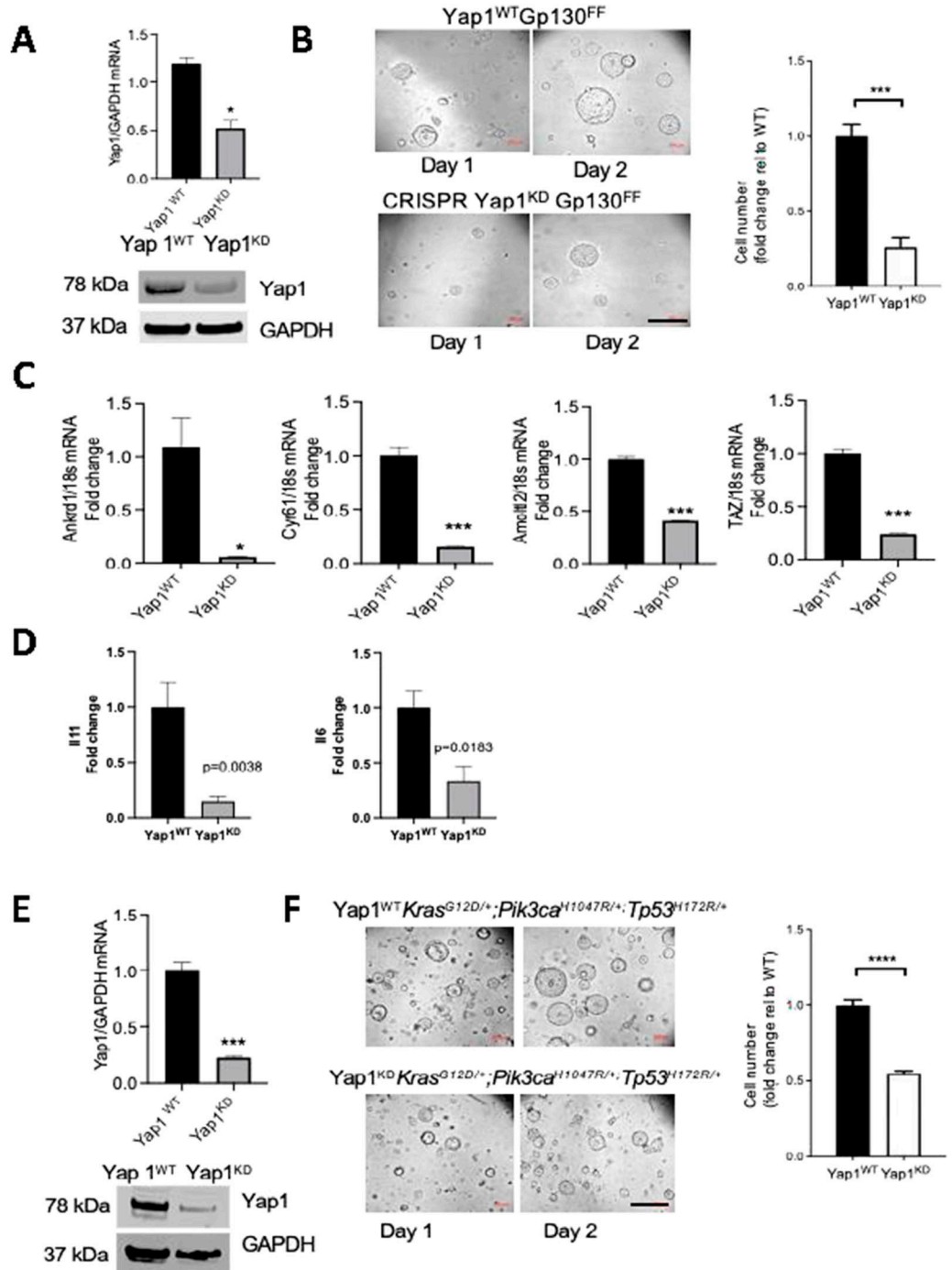

**Figure 7. CRISPR-mediated inhibition of YAP1 in gastric tumor organoids inhibits proliferation.**
**(A)** Analysis of Yap1 mRNA and protein from $Yap^{WT}Gp130^{FF}$ and Yap1 knockdown ($Yap1^{KD}Gp130^{FF}$) organoids normalised for expression of GAPDH. **(B)** Light microscopy images of $Yap^{WT}Gp130^{FF}$ and $Yap1^{KD}Gp130^{FF}$ organoids in culture and cell proliferation rates measured with the Real-Time Glo assay. Data are presented as a fold-change of luminescence compared with counts at the initiation of experiment. Scale bar = 300 $\mu$m. **(C)** Changes in mRNA transcript expression of Yap1 target genes in $Yap^{WT}Gp130^{FF}$ and $Yap1^{KD}Gp130^{FF}$ organoids measured by qRT–PCR. Expression is normalised to 18S values. **(D)** Changes in mRNA transcript expression of IL-11 and IL-6 cytokines in $Yap^{WT}Gp130^{FF}$ and $Yap1^{KD}Gp130^{FF}$ organoids measured by qRT–PCR. Expression is normalised to 18S values. **(E)** Analysis of Yap1 mRNA and protein expression in $Yap1^{WT}Kras^{G12D}Pi3kCa^{H1027R}Tp53^{R172H}$ and $Yap1^{KD}Kras^{G12D}Pi3kCa^{H1027R}Tp53^{R172H}$ gastric tumor organoids. Data were normalised for expression of GAPDH. **(F)** Light microscopy images of $Yap1^{WT}Kras^{G12D}Pi3kCa^{H1027R}Tp53^{R172H}$ and $Yap1^{KD}Kras^{G12D}Pi3kCa^{H1027R}Tp53^{R172H}$ organoids in culture and cell proliferation rates measured with the Real-Time Glo assay. Data are presented as a fold-change of luminescence compared with counts at the initiation of experiment. Scale bar = 300 $\mu$m. Data are mean ± SEM from three independent experiments performed in quadruplicate, $*P < 0.05$, $**P < 0.001$, $***P < 0.0001$, unpaired $t$ test.

(UCSC) ([47]). Gene expression values were normalized using the mean-centered Z-score method ([48]). Gene expression for *YAP1*, *TEAD1*, *IL6ST*, *IL11*, *IL11RA*, *IL6*, *IL6R*, and *STAT3* was extracted. Clinical information was obtained from cBioportal for the TCGA-STAD cohort. TME-subtypes were obtained from the recent study by reference [26]. Relevant clinical information such as histological and molecular classification of tumors and TME-subtypes were matched to sample-ID using Rstudio (v4.1.0). The visualisation of the gene expression distribution and statistical testing by multiple comparisons one-way ANOVA, matched tumor versus normal gene expression plots and paired *t* tests were performed in GraphPad Prism (v8.4.3). Node-positive gastric cancer patient survival and gene expression data correlation were performed using the Kaplan–Meier plotter online resource ([49]). Data visualisation of gene expression from matched normal and tumor tissue was performed using TNM plot ([50]).

### Single-cell RNA sequencing and bioinformatics

For high-throughput single-cell studies, the 10X Genomics Chromium kit (v2) was used for single-cell capture and cDNA preparation according to the 10x Single-Cell 3′ Protocol as previously described ([51]). Single-cell suspension of tissue obtained from WT gastric tissue and gastric tumor tissue from 12-wk-old *Gp130$^{FF}$* mice (n = 10 mice per genotype). Freshly sorted cells were pooled and manually counted and equal numbers per sample (1,000 cells/µl) were loaded for capture. Sequencing was carried out on an Illumina Nextseq 500 with a maximum of two libraries per run. Illumina output from 10X Genomics Chromium sequencing reads was processed using Cell Ranger v3.0.2. De-multiplexing, alignment to the mm10 transcriptome and unique molecular identifier collapsing to gene level were performed using cellCounts a function within Rsubread (v2.5.0) ([52]) for processing raw 10X scRNA-seq data. Cells with less than 200 genes detected or with high mitochondrial content of >20% were filtered out. Cells with >4,000 genes detected were also filtered out to minimize the occurrence of doublets. Genes that did not map to official symbols were filtered as were genes that failed to express (an expressed gene has at least one unique molecular identifier count) in at least three cells in at least one sample. After removal of unwanted cells from the dataset, the data were normalized using a global-scaling normalization method "LogNormalize" in Seurat ([53]). Dimension reduction and cell clustering were performed using functions implemented in Seurat. An unbiased cell type annotation was performed using SingleR ([54]) against ImmGen database which consists of normalized expression values of 830 microarray samples from pure populations of murine immune cells. Manual curation of *Gp130$^{FF}$* epithelial cell populations was performed within Loupe Browser 6, cells expressing high levels of mitochondrial genes were removed and following re-clustering, epithelial cell populations were identified using well-characterized gastric epithelial markers ([55]). Low-abundance cell clusters, with less than 10% of the total number of cells, were removed and the remaining populations were labeled.

### Bulk RNA sequencing and bioinformatics

Gastric antral tumors from 18-wk-old *Gp130$^{FF}$*; *Yap1$^{WT}$* and *Gp130$^{FF}$*; *Yap1$^{KO}$* mice were dissected and epithelial cells were sorted by flow cytometry for EpCAM marker. Samples from four mice of each genotype were used to extract RNA for sequencing. Libraries for whole transcriptome analysis were generated following the Illumina's TruSeq RNA v2 sample preparation protocol. Libraries were sequenced on an Illumina NextSeq 500 to produce 80 bp paired-end reads. Paired-end reads were aligned to the mouse reference genome GRCm38/mm10 using the Subread aligner ([56]). Gene level read counts were obtained by running featureCounts ([57]), a read count summarisation program within the Rsubread package and the inbuilt Rsubread annotation, that is, a modified version of NCBI RefSeq mouse (mm10) genome annotation build 38.1. Pseudo genes or genes that did not meet a counts per million reads cut-off of 0.5 in at least 6 libraries were excluded from further analysis. Read counts were converted to log$_2$-counts per million, quantile normalized, and precision weighted with the voom function of the limma package ([58], [59]). A linear model was fitted to each gene, and empirical Bayes moderated t-statistics used to assess differences in expression ([60], [61]). Genes were called differentially expressed (DE) if they achieved a false discovery rate of 5% or less. Gene Ontology (GO) terms enrichment analysis on the differentially expressed genes was performed using the goana function within the limma package ([62]). Pathway enrichment against the KEGG pathways for differentially expressed genes was performed using the kegga function also implemented in the limma package.

### CRISPR-mediated knockdown of YAP1 in tumor organoids

Established organoids were used for CRISPR-mediated genome editing to knockdown YAP1 expression. Guide crRNA targeting exon 1 of the mouse *YAP1* gene was combined in equal proportions with tracrRNA containing an ATTO550 label. This was used to create a RNP complex with Alt-R S.p. Cas9 Nuclease V3 (Integrated DNA Technologies) which was then transfected into single-cell–separated organoids using Lipofectamine CRISPR-MAX (Life Technologies) and ATTO550-positive organoid cells were FACS sorted before organoid subculture was continued. Knockdown of YAP1 at the protein level was validated with Western blot analysis.

### Organoid imaging

Parental and Yap1 CRISPR/Cas9-mutated organoids were plated into chamber slides for imaging using a Zeiss 980 confocal microscope. Z-stacked images to capture the whole 3D organoid structure were taken and analyzed with Zen Blue software for size and shape.

### Western blot analysis of tumor lysates

Whole tumor samples or organoids were lysed with RIPA buffer supplemented with cOmplete Mini protease inhibitor cocktail (Roche) and PhosSTOP phosphatase inhibitor cocktail (Roche). Protein concentration was determined using Pierce BCA assay (Thermo Fisher Scientific) and 30 µg total protein was used for Western blot analysis. Lysates were heat denatured for 10 min at 70°c with 25% NuPAGE LDS sample buffer and 10% NuPAGE reducing agent, before separation in a precast 4–12% NuPAGE Bis-Tris mini gel (Thermo Fisher Scientific) at 200 V for 30 min. Protein band sizes were identified with a Kaleidoscope Protein Ladder (Bio-Rad). Protein

was transferred to a PDFV membrane using the iBlot2 Gel Transfer system for 7 min at 20 V, before blocking for 1 h at RT with Odyssey blocking buffer (TBS) (Li-COR). Membranes were incubated overnight with primary antibody diluted to optimal concentration in 1:1 solution of Odyssey blocking buffer (TBS) and TBST at 4°c on a rotating block (Table S3). Following washes in TBST, fluorescent secondary antibody was incubated for 1 h at RT. Membranes were then washed and visualized using the Li-COR Odyssey Imaging system.

## Immunohistochemistry of gastric and tumor tissue

Whole tumor sections were fixed in 10% neutral buffered formalin for 24 h, transferred to 100% ethanol, and then processed for paraffin embedding by the Austin Anatomical Pathology department. 4 µm thick tissue sections were cut on to SuperFrost slides for further analysis. Sections were rehydrated in xylene and alcohol before antigen retrieval was performed using citrate buffer, pH 6.0, in the microwave for 15 min. Blocking of endogenous peroxidase activity was performed using hydrogen peroxide quenching ($H_2O_2$, 30%) for 20 min, following which 10% normal goat serum in TBST was used to block non-specific antibody binding for 1 h at RT. Primary antibody was incubated overnight at 4°C, and following TBST washes biotinylated secondary (1:200) was applied for 1 h at RT (Table S3). Vectastain Elite ABC HRP Kit (Vector Laboratories) was used to enhance signal detection before chromogenic staining was performed using DAB peroxidase (Dako). Sections were then counterstained using Mayer's hematoxylin, dehydrated, and mounted using DPX. Stained sections were digitally captured using Aperio slide scanner (Leica Biosystems). Cells that were stained positive for cleaved caspase 3 (Cell Signalling), CD3 (Thermo Fisher Scientific), CD8, CD4, and F480 (Abcam) antibodies were quantified using Aperio ImageScope Positive Pixel Count algorithm and expressed as a percentage of total pixels analyzed. Data are presented as mean ± SEM where each data point is representative of a value from the tumor of an individual mouse.

## Flow cytometry

Flow cytometry was performed as previously described (63). In short, tissues were cut into 1 mm pieces and digested in collagenase/dispase (Roche) and DNase I (Roche) in $Ca^{2+}$- and $Mg^{2+}$-free Hanks medium plus 5% FCS for 30 min at 37°C with gentle shaking. Samples were then vortexed for 15 s, filtered, and washed in PBS plus 5% FCS. Single-cell suspensions were blocked with FC block (Invitrogen) for 20 min at 4°C, before staining with fluorophore-conjugated primary antibodies for 20 min at 4°C in the dark (Table S3). Cells were washed twice and re-suspended in PBS supplemented with 5% FCS before analysis with either an Aria III cell sorter or BD FACS Canto.

Isotype antibodies and fluorescent-minus-one controls were used to estimate background fluorescence in combination with either compensation beads and/or unstained controls. Dead cells were detected and excluded from analysis using Sytox Blue or Fixable Viability Dye, eF506. All experiments were analyzed with FlowJo software (Version 10).

## Quantitative qRT–PCR of tumor samples

Total RNA was isolated using the RNeasy Mini Kit (QIAGEN) (Table S3). 1 µg total RNA was used to reverse transcribe cDNA using Multiscribe Reverse Transcriptase (Thermo Fisher Scientific) primed with random hexamers. YAP1 target genes were detected using SYRB green chemistry on a Viia7 qRT-PCR system (Applied Biosystems). Primers used to assess murine genes are as follows:

*Yap1* fwd (5′-GTCCTCCTTTGAGATCCCTGA-3′); *Yap1* rev (5′-TGTTGTTGTCTGATCGTTGTGAT-3′); *Taz* fwd (5′-TGCTACAGTGTCCCCA CAAC-3′); *Taz* rev (5′-TGACCGGAATTTTCACCTGT-3′); *Ctgf* fwd (5′-GTGCCAGAACGCACACTG-3′); *Ctgf* rev (5′-CCCCGGTTACACTCCAAA-3′); *Cyr61* fwd (5′-GGATCTGTGAAGTGCGTCC-3′); *Cyr61* rev (5-CTGCATTTCTTGCCCTTTTT-3′); *Ankrd1* fwd (5′-AACGGAAAAGC GAGAAACTG-3′); *Ankrd1* rev (5′-TTCAAGCTTTGATCTTTGTTC-TAGTT-3′); *Amotl2* fwd (5′-TGACTGTACCTAAGCCGAACC-3′); *Amotl2* rev (5′-GCACACACCTGCCTAGACAAT-3′); *18S* fwd (5′-CGGCTACCACATCCAAGGA-3′); *18S* rev (5′-GCTGGAATTACCGCGGCT-3′); *Igsf9* fwd (5′-CCAGCAACGTAGCCAACATCTC-3′); *Igsf9* rev (5′-CCACAA-CACCAGCCAATACAGG-3′); *Cyp2c65* fwd (5′-GGAGGAGTTTGCTGGA AGAGGA-3′); Cyp2c65 rev (5′-GCGTCTCATCTCTTTCCAGGTC-3′); *Ptgr1* fwd (5′-GACAAAGCTGCCTGTAGAGTGG-3′); *Ptgr1* rev (5′-GCTGACCATCACGGTTTCTCCA-3′); Sult1c2 fwd (5′-GCGAACCA TCATTCAACACCGAC-3′); *Sult1c2* rev (5′-CTGAGTGGGAAGATGGGTCCTT-3′); *Chit1* fwd (5′-CTCAAGACCCTGTTAGCCGTTG-3′); *Chit1* rev (5′-GGCTGACTTCACAAAGGTCTGC-3′); *Cidea* fwd (5′-GGTGGACAC AGAGGAGTTCTTTC-3′); *Cidea* rev (5′-CGAAGGTGACTCTGGC-TATTCC-3′); *Inf2* fwd (5′-TCTGGACTCCAAGAAGAGCCTG-3′); *Inf2* rev (5′-CCTCCACATCAACTTGCTGGTG-3′); *Clca1* fwd (5′-CTGCCGCTAAAGAGCTTGAGCA-3′); *Clca1* rev (5′-ATCGCCGCATTTCCT-GAGGAGA-3′).

## Quantification of cytokines with ELISA

Whole tumor lysates were prepared and quantitated as above. Mouse DuoSet IL-11 ELISA kit and Mouse DuoSet IL-6 ELISA kit (R&D Systems) (Table S3) were used to determine the concentration of IL-11 or IL-6 in whole tumor lysates. Briefly, assay plates were coated overnight with a capture antibody, before plates were blocked for 1 h and samples and standards incubated for 2 h. Following washes, plates were incubated with detection antibody for 2 h before application of a streptavidin–HRP conjugate. Colour development was initiated with the addition of a substrate solution, and then stopped with a stop solution before the plate being read at 450 nm absorbance. Cytokine concentration was determined by data interpolated from a standard curve.

## Statistical analysis

Data from experiments were analyzed for statistical analysis with *t* test or Mann–Whitney *U* test (if data were not distributed normally), $P < 0.05$ was considered significant. Statistical analysis of differences among the means of two or more groups was performed using a Tukey's multiple comparisons one-way ANOVA test. All statistical analysis was performed using GraphPad PRISM Version 9.0 software.

# Data Availabilty

Transcriptomics data that support the findings of this study are available in the Gene Expression Omnibus (GEO) database repository (GEO accession GSE197002).

## Materials request

Materials and data request should be addressed to AL Chand, M Ernst, and W Shi.

# Supplementary Information

# Acknowledgements

We acknowledge the support of the Victorian State Government Operational Infrastructure Support, the National Medical Health and Research Council (NHMRC) of Australia, Cancer Council Victoria, and the Victoria Cancer Agency. This work was supported in part by an NHMRC Program Grant 1092788 (to M Ernst) and by fellowships from the Victorian Cancer Agency 19014 (to AL Chand) and the NHMRC Fellowship 1062247 (to AL Chand).

## Author Contributions

P Thilakasiri: conceptualization, data curation, formal analysis, and writing—original draft.
RN O'Keefe: data curation, formal analysis, investigation, visualization, methodology, and writing—original draft, review, and editing.
SQ To: formal analysis, investigation, and writing—original draft.
D Chisanga: data curation, formal analysis, methodology, and writing—original draft.
MF Eissmann, RS Dmello, and BT Kile: data curation, formal analysis, and writing—original draft.
ALE Carli: formal analysis and writing—original draft.
B Duscio: data curation and writing—original draft.
D Baloyan: data curation and formal analysis.
D Williams: resources, data curation, and methodology.
J Mariadason: conceptualization and data curation.
AR Poh: data curation, investigation, and writing—original draft.
B Pal: resources, formal analysis, and methodology.
JHA Vissers: resources, data curation, and formal analysis.
KF Harvey: data curation, formal analysis, and validation.
M Buchert: data curation, formal analysis, supervision, and writing—original draft.
W Shi: resources, data curation, formal analysis, and methodology.
M Ernst: conceptualization, supervision, funding acquisition, investigation, methodology, project administration, and writing—original draft, review, and editing.
AL Chand: conceptualization, data curation, formal analysis, supervision, funding acquisition, validation, investigation, visualization, project administration, and writing—original draft, review, and editing.

## Conflict of Interest Statement

M Ernst serves on the Scientific Advisory Board of *Lassen Therapeutics* which develops anti-IL11 therapeutics.

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
