## [Reviewer comments · Life Science Alliance]

Life Science Alliance

Mechanism of cellular crosstalk in the gastric tumor microenvironment are mediated by YAP1 and STAT3

Pathum Thilakasiri, Ryan O'Keefe, Sarah To, David Chisanga, Moritz Eissmann, Annalisa Carli, Belinda Duscio, David Baloyan, Rhynelle Dmello, David Williams, John Mariadason, Ashleigh Poh, Bhupinder Pal, Benjamin Kile, Joseph Vissers, Kieran Harvey, Michael Buchert, Wei Shi, Matthias Ernst, and Ashwini Chand

DOI: <https://doi.org/10.26508/lsa.202302411>

Corresponding author(s): Matthias Ernst, Olivia Newton-John Cancer Research Institute; Matthias Ernst, Olivia Newton-John Cancer Research Institute; and Ashwini Chand, Olivia Newton-John Cancer Research Institute

Review Timeline:

Submission Date:	2023-10-02
Editorial Decision:	2023-10-03
Revision Received:	2023-10-19
Editorial Decision:	2023-10-20
Revision Received:	2023-10-29
Accepted:	2023-10-31

Transaction Report:

Please note that the manuscript was previously reviewed at another journal and the reports were taken into account in the decision-making process at *Life Science Alliance*.

Referee #1 Review

Thilakasiri et al, "Cellular crosstalk in the gastric tumor microenvironment are mediated by YAP1 and STAT3".

Authors study inhibition of Yap1 which is known to be a key component of Hippo pathway associated with poor prognosis in many cancers. They find that inhibition of Yap1 activity has anti-tumor effects in gastric tumors driven by oncogenic mutations and inflammatory cytokines. They further show that Yap1 regulates cell metabolism, proliferation and immune responses in normal and neoplastic gastric epithelium and its function may span across different subtypes of gastric cancer. They study association of Yap1 induction with gp130/stat3 driven pathway. They find that parallel activation of Yap1 and gp130/Stat3 pathways exists in Gp130FF model (previously characterized by authors) of gastric cancer. Conversely, Tff1cre-mediated (inducible) Yap1 gene knockout in the gastric epithelium of gp130FF mice reduces gastric tumors. Also, ablation of Yap1 in established tumors reduces progression and authors identified (new) transcriptional pathways downstream of Yap1 which in full or in part account for the phenotype, with some of these pathways being related to innate and adaptive immune responses. Overall, manuscript and studies are well planned and performed. Data on Yap1 ablation in epithelium is a little bit predictable but is nevertheless striking. Opportunity to delete Yap1 "early" and "late" is exciting and adds to the manuscript and data interpretation. This study further contributes to the understanding of the role of Yap1 driven pathways in GI cancers.

Several points of minor-to-moderate criticism are listed below which could help to improve the paper.

Major:

- 1) Fig1, Fig2 and elsewhere- mice should be better described, particularly Yap1 WT. What are these? Tff1-Cre negative but floxed? Cre positive but not floxed (aka WT)? Both Cre negative and not floxed? What is the breeding scheme and colony structure for KO and WT control?
- 2) Fig 4 would benefit from more detailed flow cytometry based type of analysis which would help to further analyze the subsets- for example Th1 cells vs T regs, subsets of dendritic cells which can be immunostimulatory or tumor promoting (cDC1, cDC2 etc), exhaustion of CD8 T cells, cytotoxic signature (Prf1, Gzmb, Cd107) of NK and CD8 T cells, presence of NK cells and monocytes and neutrophils separately, with some of these cells types possibly to be regulated (at least, at the recruitment level) by Yap1 in epithelium. As presented, there is increase in immune cells but there is no documentation of actual anti-tumor immunity.
- 3) So if Fig 4 suggests anti-tumor immunity, will CD8 depletion (or CD4 deletion, as authors suggested before?) or any similar approaches cancel Yap1 deletion effect and restore tumor growth?
- 4) Fig5. Data on cell autonomous role of Yap1 in organoids is strong. Is there any transcriptional or phenotypic difference then with regard to Gp130FF mutation status. Is IL-11 produce in all of these organoid cultures? To which extent transcription signature driven by Yap1 deletion on Fig 3 intersects to changes seen in Yap1 deficient organoids only (epithelial cells only). Where the most prominent (cell death, IFN pathways, proliferation, metabolism, cell death etc) signatures are coming from?

Minor:

- 1) "positively correlates to worse overall patient survival" in introduction can possibly be rephrased
- 2) Instead of 10 mg/ml concentration of Tamoxifen, actual amount going into the mouse should be provided in Methods
- 3) Fig3B- gene symbols on the right should be visible

Referee #2 Review

Dr. Chand and collaborators aim at establishing a cross-talk between YAP1 and STAT3 in gastric tumors and propose that YAP1 and STAT3 signaling mediates innate and adaptive immune responses in the tumor microenvironment. Among their objectives, the authors state (page 3) that they sought to establish whether Yap1 activity was intrinsically associated with that of gp130/Stat3 signaling-dependent carcinogenesis and to clarify the underlying mechanisms that control gastric cancer initiation and development.

The manuscript reports solid evidence that deletion of YAP1 impairs tumor development in gastric cancer animal models, some associative data claiming YAP1 activation in gastric cancer (which I find highly questionable), but lacks any mechanistic insight showing cross-talk of YAP1 and gp130/signaling.

Main criticism:

The paper does not demonstrate a functional/biological association between YAP1 and gp130/stat3; neither provides mechanistic insight concerning their cross-regulation. YAP1 activation in gastric tumors is claimed mainly based on the observation that YAP1 protein is raised, but there is no indication of increased activity (are the target genes more expressed? Is YAP1 more associated with chromatin? Is it more nuclear?) nor there is any insight about how gp130/stat3 would activate YAP1 in gastric tumors.

At face value, the manuscript shows that YAP1 is required for tumor initiation and development but provides very little evidence concerning why YAP1 is essential for cancer cells (which are the pathways or the biological processes that are controlled by YAP1 and are relevant for cancer cell?).

Some specific points:

1. From the scRNAseq (data from figure 1F), can the Author also show the expression of canonical YAP/TAZ target genes? This will support the notion that YAP is activated in tumors.
2. Transcriptional analyses were done at a single time point; thus, it is unclear how Authors can discriminate direct effects (genes regulated by YAP1) from secondary transcriptional responses triggered by YAP1 deletion. Also, the Authors should clearly state when RNA expression was profiled and when YAP was deleted; writing (see fig legend 3) that animals were 20-week-old is not really informative.
3. On page 6, the authors state: "Therefore, this association with fibrosis and immune cell infiltrate is a specific TME landscape in which treatments targeting the Hippo and STAT3 pathways could potentially enhance anti-tumor responses " I disagree with the conclusion since in the manuscript there is no association between Hippo pathway components and IF/E TME: only two Hippo pathway components were evaluated (YAP1 and TEAD1) of which only one (TEAD1) showed positive correlation. Considering that TEAD1 is necessary but not sufficient for YAP1 dependent transcription, the elevation of TEAD1 level can hardly be taken as a sign of upregulation of YAP1-dependent transcription.
4. The link connecting YAP1, IL-11, and STAT3 activity is weak, if not totally inferential. It is unclear if the reduction of IL11 causes the reduction of stat3 or whether loss of YAP1 leads to the demise of tumor cells which, as a side effect, also stops synthesizing IL11.

Minor points

1. Page 5: Authors state: "To provide additional insights into Yap1 activity, we evaluated expression patterns on Taz and Tead1, key interacting partners of the active transcriptional complex." Why did the authors limit their analysis to TEAD1? Do they have data concerning the expression of the other TEADs?
2. Figure 6 and related text, please indicate when YAP was deleted and for how long.
3. Figure 6D. Check the labeling of the WB and insert MW marker position.
4. Table 2 was not included in the material submitted to reviewers

Referee #3 Review

In this manuscript, Thilakasiri et al. used previously described homozygous gp130Y757F (gp130FF) mutant mice to investigate the role of Yap in the development of gastric cancer. By single cell RNA sequencing the authors found abundant Yap expression in normal gastric and tumor epithelial cells derived from gp130FF mice and in silico analysis of human gastric cancer samples revealed an association between Yap expression and inflammatory markers IL6, IL11, STAT3 mRNA. Conditional deletion of Yap early during tumor development or in established tumors using Tff1-CreERT mice resulted in a significant reduction of tumor burden and size, as well as diminished Stat3 phosphorylation, MAPK signaling, and IL11 expression. Transcriptional profiling of Yap mutant tumors revealed downregulation of genes associated with cell proliferation and/or apoptosis, as well as metabolic pathways. Loss of Yap in tumors also induced an antitumor response with an increase in tumor-infiltrating immune cells. Finally, Crispr-mediated knockout of Yap1 diminished the size of gp130FF and KPT gastric tumor organoids.

This study convincingly shows that Yap is a critical driver of gp130-dependent gastric cancer progression and is consistent with previous reports showing Yap as an essential downstream effector in intestinal and pancreatic tumorigenesis. The observations that Yap depletion leads to an anti-tumor immune response in a gastric tumor model provides important insight into the function of this key tissue growth regulator. However, one of the limitations of the study is that it does not attempt to investigate the mechanisms by which Yap prevents immune cell infiltration or promotes tumor cell proliferation. As noted below this could have been investigated further in the tumor organoid models described in the study. Nevertheless, this study is very interesting and highly relevant to researchers in the Hippo community and gastric cancer community. Specific comments below should be addressed before publication.

Specific comments:

In Figure 5, the authors propose that Yap promotes tumor cell proliferation but only looked at organoid size and cell number. Proliferation and apoptosis rates of tumor cells should be examined by Edu incorporation and caspase 3 staining or by other comparable means.

The data suggests that IL-11-GP130-Yap axis may drive gastric cancer growth. To further validate this model the authors should exploit their tumor organoids by treating them with either IL6 or IL11 to determine whether this augments Yap expression or proliferation of tumor organoids. Also does depletion of Yap in organoids diminish IL11 expression or is IL11 primarily expressed in immune cells?

landscape in which treatments targeting the Hippo and STAT3 pathways could potentially enhance anti-tumor responses " I disagree with the conclusion since in the manuscript there is no association between Hippo pathways components and IF/E TME: only two Hippo pathway components were evaluated (YAP1 and TEAD1) of which only one (TEAD1) showed positive correlation. Considering that TEAD1 is necessary but not sufficient for YAP1 dependent transcription, the elevation of TEAD1 level can hardly be taken as a sign of upregulation of YAP1- dependent transcription.

Authors response: We agree with the reviewer and therefore have removed this statement from the manuscript.

Reviewer response: I thank the Authors for accepting my remark.

Point 4: The link connecting YAP1, IL-11, and STAT3 activity is weak, if not totally inferential. It is unclear if the reduction of IL11 causes the reduction of stat3 or whether loss of YAP1 leads to the demise of tumor cells which, as a side effect, also stops synthesizing IL11.

Author response: We agree with the reviewer that the data in our original submission did not allow to unambiguously distinguish whether Yap1 ablation results in a reduction of IL11 (and associated Stat3 activation) or whether a Yap1-dependent demise of tumor cells reduces the number of epithelial cells that can produce IL11.

In order to clarify this issue, we have now assessed IL11 expression in organoids established from gp130FF tumors in which we also ablated Yap1 expression by CRISPR gene editing (Figure 6). Our analysis suggested that on a per cell basis, Yap1-knockdown organoids express less IL11 than their Yap1-proficient counterparts (new Figure 7G). We therefore conclude that suppression of Yap1 signaling in gastric epithelial cells reduces their IL11 expression. In turn this is with the reduced IL-11 levels measured by ELISA in the tumors of gp130FF;Yap1KO mice.

Reviewer response: I think that despite the new data showing that the loss of YAP1 in organoids leads to reduction of IL-11, the main question concerning how YAP1, IL-11 and STAT3 are linked is still not addressed. This data suggests that IL-11 drop is a consequence of the loss of YAP1, but does not clarify whether IL-11 loss is sufficient to impair tumors and whether IL-11 is needed to activate STAT3. Or more in general does not offer clues concerning whether and how YAP1,IL-11 and STAT3 are part of a regulatory network that may support tumor growth and interaction with the microenvironment.

Minor points were addressed by the Authors.

Referee #3 Review

Report for Author:

The revised version of the manuscript addresses my specific comments by showing Yap-drives IL11 expression in GP130 driven tumor organoids. Despite this, the conclusions at the end of the results section on page 14 are unjustified:

"Collectively, our results obtained across gp130FF and KPT tumor organoids and Yap1 mutant gastric tumor organoids suggests that inhibition of Yap1 activity confers suppression of tumors via an IL11-mediated, tumor-cell intrinsic mechanism that possibly is further augmented by an enhanced anti-tumor immune response."

Although it is clear that tumor organoid growth is dependent on Yap, the authors have yet to show that this is via an IL-11 mediated , tumor cell intrinsic mechanisms. As far as I could tell previous reports by the group used whole body knockouts and therapeutic approaches to deplete IL-11 signaling. Thus it remains unclear what direct effects IL-11 has on tumor cells. One would assume that IL-11 treatment of GP130 organoids would have no effect, but it would have been interesting to test whether IL-11 treatment augments growth of KPT or wt organoids as I previously requested. Moreover, the authors did not check whether knockout of Yap depletes IL-11 levels in KPT organoids. Therefore, there is very little evidence to suggest that tumor cell derived IL-11 acts in an autocrine function to promote tumor proliferation and survival.

Overall, the reviewer recognizes the improvements made in the manuscript but some of the conclusions remain problematic. Therefore, I suggest that the above statement and the discussion be revised to reflect the uncertainty about the mechanism of action of IL-11.

Referee #1 Review

Report for Author:

The paper has improved and in my view is acceptable, the only thing which I believe needs to be modified/changed is around tamoxifen treatments:

- 1) some details are now in Fig legends, but also in Materials and Methods it should be written that tamoxifen is injected to induce the knockout, but the control is without tamoxifen (not without Cre).
- 2) For some experiments/figures it is OK that tamoxifen was injected/treated with and then stopped and mice collected; however in some other experiments tamoxifen treatment goes all the way to the collection point. Here at least a possibility that some changes which authors see may be due to tamoxifen/no tamoxifen treatment and not due to KO/WT conditions should be acknowledged and discussed in discussion.

Referee #2 Review

Report for Author:

I'll start by thanking the Authors for taking the time to revise the manuscript.

After reading the revised version and the Authors' responses, I am afraid I still have concerns regarding many of the key claims of this manuscript.

The manuscript title states "Mechanisms of cellular crosstalk in the gastric tumor microenvironment are mediated by YAP1 and STAT3" and indeed the Authors show that upon YAP1 deletion there is a change in the immune infiltrate, which is associated with IL-11 reduction. Yet, there is no mechanistic insight on how IL-11 may control the immune infiltrate in these tumors, nor there are indications of whether IL-11 is a direct target of YAP1, or whether IL-11 expression is regulated by other transcription

factors (NF- κ B?, see DOI: 10.1186/s10020-019-0103-4).

In addition, the role of STAT3 is left unexplored (despite being mentioned in the title).

Thus, I do not see a strong mechanistic insight suggesting how YAP1 or STAT3 contribute to this crosstalk.

Whether the remodeling of the tumor microenvironment following YAP1 deletion contributes to tumor regression is also left unaddressed, thus the closing sentence on the abstract saying "We propose that the Hippo pathway is targetable across gastric cancer subtypes and its therapeutic benefits are mediated by regulating cellular crosstalk in the tumor microenvironment." is not supported by any experimental evidence.

The genetic evidence showing the requirement of YAP1 in tumors, both in-vivo and in-vitro, is solid: it indicates a strong YAP1-dependence but lacks in mechanistic insight. Why are tumors and organoids failing? Is it due to loss of IL-11 expression? How is STAT3 activated by YAP1? Is STAT3 modulation a cause or a consequence of the demise of tumors following YAP1 ablation? And more in general, what are YAP1 and STAT3 roles in tumor cells? What key processes are controlled by YAP1 and STAT3?

So in summary, many of the main claims of this manuscript do not seem to be supported by experimental evidence, but rather are speculative.

Point by point rebuttal to Authors response

Point 1: The paper does not demonstrate a functional/biological association between YAP1 and gp130/stat3; neither provides mechanistic insight concerning their cross-regulation. YAP1 activation in gastric tumors is claimed mainly based on the observation that YAP1 protein is raised, but there is no indication of increased activity (are the target genes more expressed? Is YAP1 more associated with chromatin? Is it more nuclear?) nor there is any insight about how gp130/stat3 would activate YAP1 in gastric tumors. At face value, the manuscript shows that YAP1 is required for tumor initiation and development but provides very little evidence concerning why YAP1 is essential for cancer cells (which are the pathways or the biological processes that are controlled by YAP1 and are relevant for cancer cell?).

Authors response: As per the reviewer's suggestion, we quantified nuclear and cytoplasmic localisation of Yap1 by immunohistochemistry. The increase in nuclear localisation of Yap1 in tumors with increasing age of the mice suggests that the tumor-promoting role of Yap1 is associated with its excessive transcriptional activity (Figure 1C and 1E). Our single cell RNA analysis of gastric tumors for genes that are prototypical for a Yap1 signature (Figures 2A, 4C), suggests that Yap1 controls critical tumor-promoting pathways. Indeed, we find that the altered expression of these genes also occurs following genetic manipulation of Yap1 expression in organoids (Figure 7F, H). Most importantly, selective ablation of Yap1 expression in the glandular epithelium, which comprises the majority of cells in gastric tumors of gp130FF mice, reduces proliferation of the neoplastic epithelium while simultaneously stimulating apoptosis (Fig 4C-I). Collectively, these observations provide a compelling argument for a functional biological association between Yap1 and gp130/Stat3 signaling.

Reviewer response: the Authors response addresses only some of the points raised (they now provide stronger evidence for YAP1 activation in tumors) but leaves unaddressed some key issues (how is YAP1 activated in tumors? Cross regulation of YAP1 and STAT3? Mechanistic insight on how YAP1 support tumor cell fitness).

Point 2: Transcriptional analyses were done at a single time point; thus, it is unclear how Authors can discriminate direct effects (genes regulated by YAP1) from secondary transcriptional responses triggered by YAP1 deletion. Also, the Authors should clearly state when RNA expression was profiled and when YAP was deleted; writing (see fig legend 3) that animals were 20-week-old is not really informative.

Authors response: The reviewer is correct that our analysis does not allow us to discriminate between direct/primary, and indirect transcriptional responses to Yap1 ablation in vivo. Indeed, given pharmacokinetic and pharmacodynamic consideration of tamoxifen-induction of Cre activity and subsequent recombinase-dependent Yap1 gene ablation in vivo, it is not possible to unambiguously distinguish between direct/primary and subsequent changes to gene expression profiles. This is further compounded by observations that Cre recombinase activity is highest during the S-phase of the cell cycle (Hashimoto, J.Biochem 2008), when only a minority of gastric epithelial cells in gp130FF tumors is proliferating in a non-synchronized manner. We therefore elected to perform our analysis one days after tamoxifen administration as a compromise between maximizing the number of transcriptomes analysed from Cre-recombined cells, and minimizing the timeframe in which transcripts from recombined cells accumulate. We believe this to be a valuable approach, because in this setting we observe changes to expression of target genes of the canonical Yap1 signaling pathway.

To provide clarity about the time frame in which we analyzed gene expression, we have updated the Methods Section; the legend to Figure 4 now clearly states that Tamoxifen was administered for 3 days to 20-week-old animals, and tumor collected the day after tamoxifen treatment ended.

Reviewer response: Given the clarification of how the experiments were performed, I agree with the arguments put forward by Authors.

Point 3: On page 6, the authors state: "Therefore, this association with fibrosis and immune cell infiltrate is a specific TME

In Figure 3 the authors transcriptionally profiled Yap mutant gastric tumors and uncovered several differentially expressed pathways. To determine whether these are cell autonomous changes, the authors should validate some these genes in their organoid model. This should provide insight into which biological processes Yap directly regulates as opposed to secondary effects of Yap depletion.

Minor comments:

Page 4: "We show that Yap1 drives drive gastric epithelial..."

The authors should describe in more detail the methodology used to create Yap KO organoids. To my knowledge this is the first time that a CRISPR based protein delivery system has been used to knockout a gene of interest in organoids. This is an interesting methodology that merits more information. Also, it would be informative to see immunofluorescence stainings of Yap on KO organoids to better assess targeting efficiency.

October 3, 2023

Re: Life Science Alliance manuscript #LSA-2023-02411-T

Dr. Ashwini L. Chand
Olivia Newton-John Cancer Research Institute
Cancer and Inflammation
Level 5, ONJ Centre, 145 Studley Road
Heidelberg, VIC 3084

Dear Dr. Chand,

Thank you for submitting your manuscript entitled "Mechanisms of cellular crosstalk in the gastric tumor microenvironment are mediated by YAP1/STAT3" to Life Science Alliance. We invite you to submit a revised manuscript addressing the following Reviewer comments:

- Address Reviewer 1's comments.
- Address Reviewer 2 and 3's comments by toning down claims in order to reflect the concerns regarding the limited evidence into the regulation and mechanism of action of IL-11.

Thank you for this interesting contribution to Life Science Alliance. We are looking forward to receiving your revised manuscript.

Sincerely,

- A letter addressing the reviewers' comments point by point.
- An editable version of the final text (.DOC or .DOCX) is needed for copyediting (no PDFs).
- High-resolution figure, supplementary figure and video files uploaded as individual files: See our detailed guidelines for preparing your production-ready images, <https://www.life-science-alliance.org/authors>
- Summary blurb (enter in submission system): A short text summarizing in a single sentence the study (max. 200 characters including spaces). This text is used in conjunction with the titles of papers, hence should be informative and complementary to the title and running title. It should describe the context and significance of the findings for a general readership; it should be written in the present tense and refer to the work in the third person. Author names should not be mentioned.
- By submitting a revision, you attest that you are aware of our payment policies found here: <https://www.life-science-alliance.org/copyright-license-fee>

B. MANUSCRIPT ORGANIZATION AND FORMATTING:

October 16, 2023

Editor, Life Science Alliance

Dear Dr Sawey,

We thank the reviewers for their additional comments which we address point by point in the below section.

As a general request summarized by you and the editor of another journal in response to the additional comments raised by the reviewers, we were asked to *“tone down” our claims regarding the involvement of IL11.*

Response: We have substantially amended our statements on in the summary paragraph of the Discussion section (page 19)

Specific comments:

Referee #1:

The paper has improved and in my view is acceptable, the only thing which I believe needs to be modified/changed is around tamoxifen treatments:

1) some details are now in Fig legends, but also in Materials and Methods it should be written that tamoxifen is injected to induce the knockout, but the control is without tamoxifen (not without Cre).

Response: We have now clarified this important point in the Material and Methods section (page 20).

2) For some experiments/figures it is OK that tamoxifen was injected/treated with and then stopped and mice collected; however in some other experiments tamoxifen treatment goes all the way to the collection point. Here at least a possibility that some changes which authors see may be due to tamoxifen/no tamoxifen treatment and not due to KO/WT conditions should be acknowledged and discussed in discussion.

Response: This is an important and deliberate decision we have taken to frame slightly different questions as the story develops from one Figure to the next. However, we agree that we have not explained this in the manuscript in a satisfactory way and possibly further confused the issue with the schematics that imply that Tamoxifen-treatment was for a prolonged time, rather than the *once off over a period of 3-days* as clearly stated in the Methods section. Thus, we propose to rectify this by re-ordering the initial figures (without adding or deleting any data from the previous submission) to be better in line with the logic underpinning the different regimes of tamoxifen administration:

- Revised Figure 1 and Supplemental Figure 1 demonstrates in **mice** correlation between excessive IL11/gp130/Stat3 signaling in gp130FF mice and activity of the Yap pathway
 - Main Figure – Show panels Fig 1A-1H of the previous version
 - Suppl Figure – Show panel Fig S1A-S1B of previous version AND Panel Fig 2A from previous version

- Revised Figure 2 and Supplemental Figure 2 demonstrates in **human** gastric cancer patients a correlation between excessive IL11/gp130/Stat3 signalling in gp130FF mice and activity of the Yap pathway.
 - Main Figure – Show panels Fig 1I AND Fig 3G of the previous version
- Revised Figure 3 and Supplemental Figure 3 demonstrates in mice the causal involvement of Yap for tumour growth resulting in from excessive IL11/gp130/Stat3 signalling in gp130FF mice.
 - Main Figure – Show panels Fig 3A-3F of the previous version
 - Suppl Figure – Show panels Fig 2A-2B of previous version
- All other figures remain unchanged

We first establish whether ablation of YAP by tamoxifen results in an observable on **established tumours** when analyzed at 20 weeks of age – For this we administer Tamoxifen at 16 weeks, because we have previously observed in the context of Tff1:CreERT2-mediated ablation of Stat3 in tumours that a 4-week follow-up period after Cre activation is the minimal amount of time to detect significant reduction in tumour burden. Thus, we administer tamoxifen at week 16 and perform all analysis 4weeks later (old panels 3A-3F).

We then establish that Yap ablation has also an effect during the **emergence of tumours**. For this we administer Tamoxifen at week 9 and perform analysis of tumour burden 4-weeks later (old panels 2B-2C).

Finally, for all the changes **affected gene expression**, we analyse samples immediately following tamoxifen-dependent activation of Cre-recombinase (i.e. Fig 4, etc)

Referee #2:

The manuscript title states "Mechanisms of cellular crosstalk in the gastric tumor microenvironment are mediated by YAP1 and STAT3" and indeed the Authors show that upon YAP1 deletion there is a change in the immune infiltrate, which is associated with IL-11 reduction. Yet, there is no mechanistic insight on how IL-11 may control the immune infiltrate in these tumors, nor there are indications of whether IL-11 is a direct target of YAP1, or whether IL-11 expression is regulated by other transcription factors (NF-kB?, see DOI: 10.1186/s10020-019-0103-4).

In addition, the role of STAT3 is left unexplored (despite being mentioned in the title).

Thus, I do not see a strong mechanistic insight suggesting how YAP1 or STAT3 contribute to this crosstalk.

Response: The reviewer is correct that we have not defined the exact mechanism by which Yap-signaling regulates IL11 expression in epithelial cells, which we find reduced following Yap knock-down in organoids established from tumours of our gp130FF mice. However, the reviewer is inaccurate with respect to the comment on Stat3. We have previously shown by conditional genetic Stat3 ablation as well as pharmacological inhibition of Stat3 that either treatment reduces the tumour burden in gp130FF mice (Ernst *et al.*, J Clin Invest 2008, Stuart *et al.*, Mol Cancer Therapeutics 2014). *We now refer explicitly to these papers in the 4th paragraph of the Introduction section (page 5) and the last paragraph of the Discussion section (page 19).*

With this in mind, our observations may be based on two (not mutually exclusive) mechanisms linking Yap signalling to excessive Stat3 activity and thus tumour growth: **(1)** A NF-kB dependent pathway by which YAP activates IL11 gene transcription as described by Wang *et al.*, Mol Med (2019), **(2)** A direct interaction between Yap and (unphosphorylated) Stat3 to form a transcriptionally active, heterodimeric complex as He *et al.*, Circ Res (2018). In light of the tight

correlation between Yap signalling and the abundance of IL11 and pStat3, the latter mechanism seems less likely. *We now refer explicitly include refer to these possibilities in the last paragraph of the Discussion section (page 19).*

The reviewer is also correct that we do not provide a definitive causal mechanism by which IL11 in the gp130FF model controls anti-tumour immune response. However, and as already previously outlined in the Discussion section, we have shown that IL11 responsiveness of host cells suppresses anti-tumour immune responses and that this effect is lost upon antibody-mediated CD4, but *not* CD8 cell ablation. This observation is entirely consistent with the observation presented here that upon Yap-ablation we not only find reduced IL11 expression but also increased activity of CD4, but *not* CD8 cells.

Whether the remodeling of the tumor microenvironment following YAP1 deletion contributes to tumor regression is also left unaddressed, thus the closing sentence on the abstract saying "We propose that the Hippo pathway is targetable across gastric cancer subtypes and its therapeutic benefits are mediated by regulating cellular crosstalk in the tumor microenvironment."

Response: We agree with the reviewer that our statement in the abstract was too over-reaching. We now rephrase this sentence as follows:

"We propose that the Hippo pathway is targetable across gastric cancer subtypes and its therapeutic benefits are likely to be mediated by both cancer cell-intrinsic and extrinsic mechanisms".

The genetic evidence showing the requirement of YAP1 in tumors, both in-vivo and in-vitro, is solid: it indicates a strong YAP1-dependence but lacks in mechanistic insight. Why are tumors and organoids failing? Is it due to loss of IL-11 expression? How is STAT3 activated by YAP1? Is STAT3 modulation a cause or a consequence of the demise of tumors following YAP1 ablation? And more in general, what are YAP1 and STAT3 roles in tumor cells? What key processes are controlled by YAP1 and STAT3?

Response: We do not agree with the reviewer's interpretation that tumour are "failing", but rather that Yap-deficient tumours net growth is decrease. In part this is due to tumour-intrinsic mechanisms (we note that YAP-knockdown organoids still grow in culture; Fig 6B), we also know that organoids established from gp130-wildtype mice grow less well than from gp130FF tumours (Eissmann *et al.*, Nature Comms 2019). The latter observation implies that **(1)** Stat3 activity is a determinant of organoid growth, **(2)** excessive growth of gp130FF organoids requires the presence of the gp130 receptor ligand IL11 (or another cytokine of the gp130 family); **(3)** Stat3 modulation is a cause, rather than a consequence of the "demise of tumours".

Finally, we would like to point out to the reviewer the extensive literature available on the roles of Stat3 and YAP signalling. Therefore, we believe that reviewing the "key processes controlled by Stat3 and YAP" fall well beyond the scope of our manuscript.

We confirm that none of the findings in the revised manuscript are published nor currently under consideration at any other journal.

We look forward to your comments and suggestions.

Sincerely,

October 20, 2023

RE: Life Science Alliance Manuscript #LSA-2023-02411-TR

Dr. Matthias Ernst
Olivia Newton-John Cancer Research Institute
ONJ Centre
145 Studley Rf
Melbourne 3084

Dear Dr. Ernst,

Thank you for submitting your revised manuscript entitled "Mechanisms of cellular crosstalk in the gastric tumor microenvironment are mediated by YAP1 and STAT3". We would be happy to publish your paper in Life Science Alliance pending final revisions necessary to meet our formatting guidelines.

- please upload your main and supplementary figures as single files
- Supplementary Table S2 seems to be missing; please upload
- please add ORCID ID for the corresponding author -- you should have received instructions on how to do so
- please add your main, supplementary figure, and table legends to the main manuscript text after the references section
- please add callouts for Figures 3H and S4A-B to your main manuscript text
- please upload one figure per file

Figure Checks:

- please indicate the scale bar sizes in the Legends for Figures 4G, 4I, 5D, 7B, 7F

A. FINAL FILES:

B. MANUSCRIPT ORGANIZATION AND FORMATTING:

Sincerely,

October 31, 2023

RE: Life Science Alliance Manuscript #LSA-2023-02411-TRR

Dr. Matthias Ernst
Olivia Newton-John Cancer Research Institute
ONJ Centre
145 Studley Rf
Melbourne 3084
Australia

Dear Dr. Ernst,

Thank you for submitting your Research Article entitled "Mechanism of cellular crosstalk in the gastric tumor microenvironment are mediated by YAP1 and STAT3". It is a pleasure to let you know that your manuscript is now accepted for publication in Life Science Alliance. Congratulations on this interesting work.

DISTRIBUTION OF MATERIALS:

Again, congratulations on a very nice paper. I hope you found the review process to be constructive and are pleased with how the manuscript was handled editorially. We look forward to future exciting submissions from your lab.

Sincerely,
